# Perforin-2 clockwise hand-over-hand pre-pore to pore transition mechanism

Fang Jiao [1,2,3] ✉, François Dehez[4], Tao Ni [5], Xiulian Yu[5,6], Jeremy S. Dittman [7], Robert Gilbert[5,6], Christophe Chipot[4,8] & Simon Scheuring [1,2,9] ✉

Perforin-2 (PFN2, MPEG1) is a pore-forming protein that acts as a first line of defense in the mammalian immune system, rapidly killing engulfed microbes within the phagolysosome in macrophages. PFN2 self-assembles into hexadecameric pre-pore rings that transition upon acidification into pores damaging target cell membranes. Here, using high-speed atomic force microscopy (HS-AFM) imaging and line-scanning and molecular dynamics simulation, we elucidate PFN2 pre-pore to pore transition pathways and dynamics. Upon acidification, the pre-pore rings (pre-pore-I) display frequent, $1.8 \, s^{-1}$, ring-opening dynamics that eventually, $0.2 \, s^{-1}$, initiate transition into an intermediate, short-lived, ~75 ms, pre-pore-II state, inducing a clockwise pre-pore-I to pre-pore-II propagation. Concomitantly, the first pre-pore-II subunit, undergoes a major conformational change to the pore state that propagates also clockwise at a rate ~$15 \, s^{-1}$. Thus, the pre-pore to pore transition is a clockwise hand-over-hand mechanism that is accomplished within ~1.3 s. Our findings suggest a clockwise mechanism of membrane insertion that with variations may be general for the MACPF/CDC superfamily.

The membrane attack complex (MAC)/perforin (PF)/cholesterol-dependent cytolysin (MACPF/CDC) proteins[1–4] are a large superfamily of pore forming proteins known to be crucial effectors in the immune system to fight bacterial pathogenesis. The primary function of MACPF/CDC proteins is to form transmembrane pores in membranes, with various downstream biological effects including devastation of their solute homeostasis and transport of secondary factors promoting target cell death. This superfamily is found across all kingdoms of life and is involved in diverse processes including toxic attack[5], pathogen invasion[6], development[7], immune defense[8], and inflammation[9–11]. Currently, the MACPF/CDC family is predicted to comprise over 1000 proteins[12].

These proteins characteristically undergo large conformational changes to convert from water-soluble monomers into oligomeric transmembrane pores. They use a common multistep mechanism for pore formation, despite variability in the structure and properties of the final transmembrane pore[13]. Usually this includes (i) association of a water-soluble monomer to a lipid membrane; (ii) oligomerization into pre-pore rings on top of the membrane; and (iii) transition from the pre-pore to the pore state: unraveling of two α-helical regions that refold into transmembrane hairpins (TMH1 and TMH2) in each protomer that form in the ensemble complex giant β-barrel transmembrane pores[14]. Not all MACPF/CDCs assemble into complete rings, as arcs and

[1]Department of Anesthesiology, Weill Cornell Medicine, New York City, NY, USA. [2]Department of Physiology and Biophysics, Weill Cornell Medicine, New York City, NY, USA. [3]Laboratory of Soft Matter Physics, Institute of Physics, Chinese Academy of Sciences, Beijing, China. [4]Laboratoire International Associé, Centre National de la Recherche Scientifique et University of Illinois at Urbana-Champaign, Unité Mixte de Recherche no 7019, Université de Lorraine, Vandœuvre-lès-Nancy cedex, France. [5]Division of Structural Biology, Wellcome Centre for Human Genetics, University of Oxford, Roosevelt Drive, Oxford, UK. [6]Calleva Research Centre for Evolution and Human Sciences, Magdalen College, University of Oxford, Oxford, UK. [7]Department of Biochemistry, Weill Cornell Medicine, New York, NY, USA. [8]Department of Physics, University of Illinois at Urbana-Champaign, Urbana, IL, USA. [9]Kavli Institute at Cornell for Nanoscale Science, Cornell University, Ithaca, New York, USA. ✉e-mail: fang.jiao@iphy.ac.cn; sis2019@med.cornell.edu

incomplete rings have been shown to perforate membranes in many cases[14–29].

Perforin-2 (PFN2, MPEG1) sits in an evolutionarily ancient arm of the MACPF/CDC phylogeny[30], and facilitates destruction of phagocytosed bacteria through the formation of pores in their cell membranes. PFN2 has recently been identified as a critical component in innate immunity conserved throughout the animal kingdom[31–34] and evolutionarily close to the last common ancestor of MAC and perforin-1[34, 35]. PFN2 is a crucial component of host defense against a wide spectrum of infectious bacteria, as shown in both mice and humans[36]. Knockdown or loss of PFN2 renders individual phagocytes and whole organisms vulnerable to bacterial pathogens. Recent studies showed that PFN2 was not only constitutively expressed in macrophages and permanently functional in phagolysosomes as a pore-forming protein[36–38], but it was also detected in murine embryonic fibroblasts (MEF) and human epithelial cells after bacterial infection[35, 39].

Despite extensive studies of the MACPF/CDC pore formation process[14–29, 40–44] and recent advances in the determination of the pre-pore and pore state structures of PFN2[44, 45] and MAC[41, 46] using cryo-electron microscopy (cryo-EM), the detailed mechanism of the MACPF/CDC pre-pore to pore transition pathway and its dynamics remain largely unknown[47]. Previous studies suggest PFN2 can form pores on time scales of one second[44, 45]. Thus, here we used high-speed atomic force microscopy (HS-AFM) to acquire further details of the PFN2 pre-pore to pore transition pathways and dynamics. HS-AFM is a powerful tool for the investigation of biomolecular conformational changes at high spatiotemporal resolution in physiological buffer and ambient temperature and pressure[48–54]. Recording movies at 200 ms temporal resolution of the pre-pore to pore transition of PFN2 oligomers, we uncover a multi-step, clockwise (CW) conformational transition, in which (i) pre-pore rings break, (ii) the subunit located CW from the breakage site undergoes a conformational change from pre-pore-I to a novel pre-pore-II state, (iii) this conformational transition is propagated further CW, while (iv) the initiating subunit undergoes a further conformational change to the pore state, that (v) is also propagated CW. Thus, the pre-pore to pore transition is a processive clockwise hand-over-hand process, in which the conformational transition of a single subunit induces the sequential and directional transition of distal subunits. Given the evolutionary proximity of PFN2 to the common origin of all MACPF/CDC proteins and the structural resemblance between the proteins in this family, it is likely that aspects of this previously unobserved mechanism may be conserved throughout the entire protein family.

## Results

### PFN2 pre-pore and pore structure analysis
The PFN2 ectodomain features the MACPF family domain, including the two transmembrane hairpins (TMH1, TMH2), at the N terminus of the protein, followed by an epidermal growth factor (EGF)-like repeat, the membrane-targeting P2 (also termed MABP) domain, and finally a C-terminal tail (CTT) (Fig. 1a). PFN2 assembles into a 16-subunit-ring pre-pore complex as depicted in a previously solved structure (Fig. 1b, c)[44, 45]. The PFN2 pre-pore forms an ~8 nm thick doughnut-like structure with the MACPF and P2 domains arranged concentrically, from which the MACPF domain also protrudes upwards ~1 nm. Viewing a single subunit from the side (ring cross section), reveals the MACPF domain is located towards the ring center, and the P2 domain towards the periphery, linked by a truncated EGF-like domain (Fig. 1c). The interface between subunits is created by one monomer wedging against the back face of its neighbor by forming β sheets both at the interface between MACPF domains and through association of the CTT of one subunit with the P2 domain of the next subunit[44]. There is charge complementarity between the interfaces, which is likely important for the H$^+$-dependent activation process (further analyzed below). There are two sets of transmembrane hairpin (TMH1 and TMH2) α-helices within the

MACPF domain that ultimately form a β-sheet traversing the target membrane upon pore formation (Fig. 1c, g). The CTT domain of PFN2 is unique and not present in other MACPF/CDC superfamily proteins (Fig. 1a). It packs against the P2 domains of adjacent subunits to create a continuous intermolecular β-sheet to further strengthen the assembly stability and faces the target membrane in the pre-pore structure (Fig. 1c). The CTT domain was not resolved in the PFN2 pore (Fig. 1g).

When PFN2 monomers were incubated on a freshly cleaved mica surface at pH 7.5, a wide distribution of arcs as well as rings were observed by HS-AFM (Fig. 1d, k). The pre-pores had a thickness of ~8 nm, in good agreement with the pre-pore structure determined by cryo-EM (Fig. 1j, see Fig. 1b). In contrast, on a *E. coli* lipid membrane, PFN2 formed almost entirely complete pre-pore rings (>90%) (Fig. 1e, m). The PFN2 pre-pore rings arranged hexagonally on the lipid membrane (Fig. 1e). The fact that pre-pore PFN2 forms mainly fragmental arcs on mica while it almost exclusively forms complete rings on the membrane indicates that two-dimensional (2D) lateral diffusion−only permitted on the membrane−is crucial for protomer encounter and full ring formation by PFN2.

In our previous report[44], we found that acidic pH triggered PFN2 pre-pore to pore transition. The pre-pore and pore structures revealed a 180° rotation of the MACPF domain (around an axis parallel to the membrane) with respect to the P2 domain. Other conformational changes accompany membrane insertion, including the deployment of TMH1 and TMH2 to form a 4-stranded β-sheet. These TMHs from all subunits build together the large β-barrel pore (Fig. 1b, f, g). Thus, we also characterized the PFN2 pore state using HS-AFM, first preparing the pre-pores at pH 7.5, then decreasing the pH to pH 4.0 on mica (Fig. 1h). In line with the pre-pores on mica, PFN2 pores showed a wide distribution of arcs and rings on mica (Fig. 1h, k). The height of the pore structures on mica was ~14 nm in acceptable agreement with the structure (Fig. 1h, j, see Fig. 1f). Acidification of pre-pores rings on the *E. coli* lipid membrane resulted in PFN2 membrane pores, accompanied by two major structural changes: First, while PFN2 pre-pores had ~8 nm height above mica (Fig. 1j) and membrane (Fig. 1l), the overall height of PFN2 pores was ~14 nm on mica (Fig. 1j) and ~16 nm in the lipid membrane, ~12 nm protruding out of the lipid bilayer plus ~4 nm accounting for the lipid bilayer thickness and the solution layer below (Fig. 1l, see Fig. 1f). The full height of the pores of ~16 nm down to the mica surface could be measured directly next to a membrane defect (Supplementary Fig. 1). Thus, the conformational transition is manifested by a ~4 nm increase in height during the pre-pore-to-pore transition on a lipid membrane (Fig. 1l). Second, while >90% of the pre-pore assemblies on a lipid membrane were complete rings only ~4% of the pores were complete rings and the majority of the pores were arcs constituted by 5–20 (peaking at 12) subunits (Fig. 1m).

### Clockwise PFN2 pre-pore to pore transition
The PFN2 structures and HS-AFM results of the PFN2 pre-pore and pore states suggest that the transition from pre-pore to pore requires at least three significant steps: (i) Pore formation must begin with a breaking of the interface between two pre-pore subunits (given the wide distribution of PFN2 pore arcs rather than rings), (ii) flipping of the MACPF domain by 180° with respect to the P2 domain, and (iii) unfurling of the TMHs into β hairpins entering the target membrane. The second step is related to the first, because flipping the MACPF domain in one subunit necessarily disrupts its interaction with the neighboring subunits. The second step is described here within the framework of the so-called cis-model, which suggests that the MACPF domain flips and brings its TMHs into proximity of the membrane on which the pre-pore sits and into which the TMHs unfurl to create the pore[44, 45]. An alternative model, the so-called trans-model, has been proposed according to which the pore forming TMHs are extended into the opposite direction with respect to the membrane on which the pre-pore sits attacking a second target membrane[44, 45]. We prefer the cis-model for the interpretation of

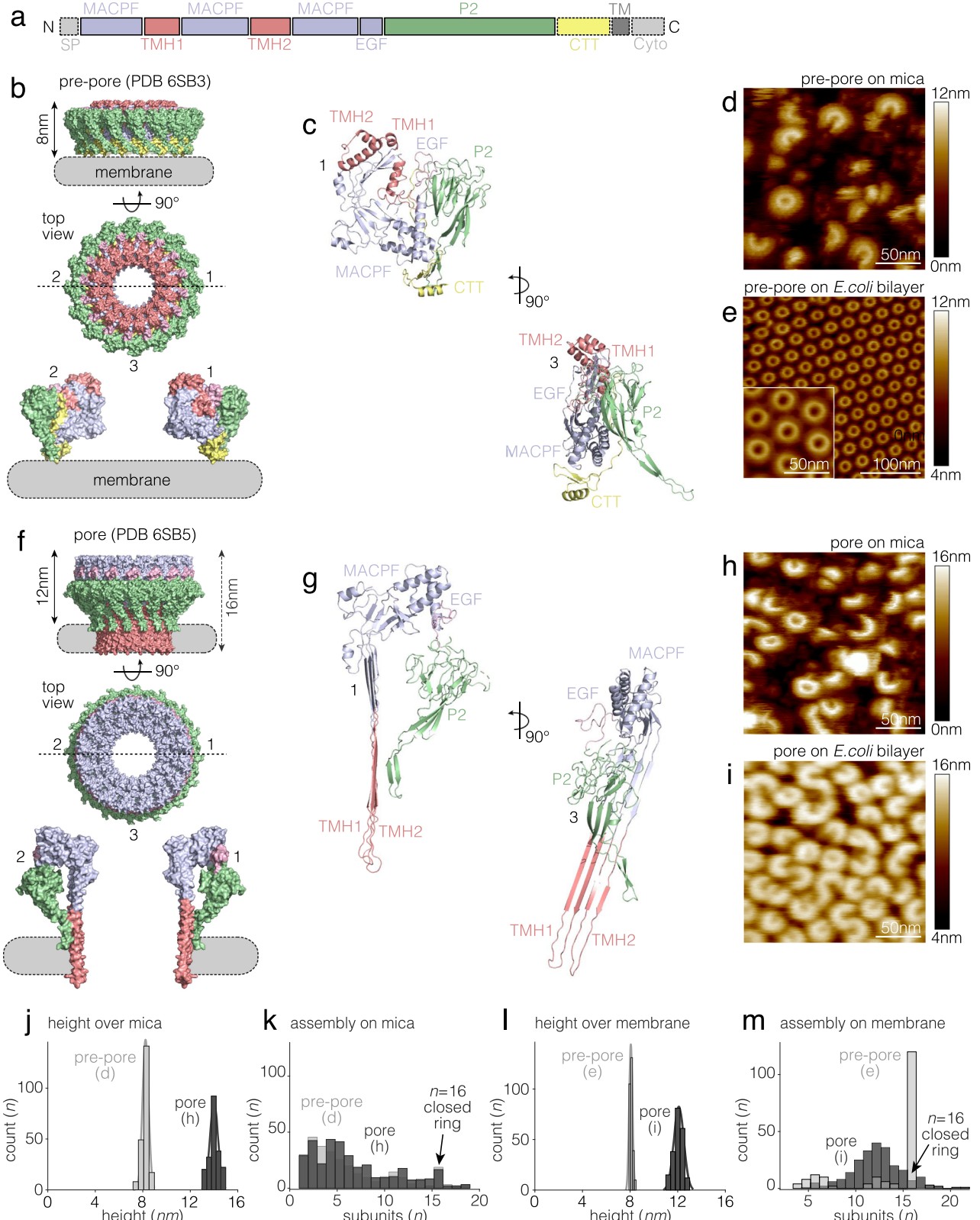

our results for its simplicity, because it agrees with the available structure of the PFN2 pore[44], and because the trans-model would implicate that a pore β-barrel is formed into the bulk solution in our experimental setup, which seems highly unlikely.

Thus, we set out to investigate the PFN2 pre-pore to pore transition mechanism and its dynamics in further detail, performing direct pre-pore to pore transformation experiments in the HS-AFM fluid cell

at room temperature, while imaging at 200 ms temporal resolution. PFN2 pre-pore rings were allowed to assemble in a dense hexagonal packing on the membrane at pH 7.5. The buffer pH was then decreased to pH 4.0. At pH 4.0, pre-pore rings showed dynamic lateral openings between subunits (Fig. 2a, arrowheads in $t = -1.0$ s to $t = 0.0$ s). Such pre-pore inter-subunit openings at low pH occurred at a rate of least ~1.8 s$^{-1}$ (Fig. 2b, see Methods), and never persisted for longer than one

**Fig. 1 | PFN2 pre-pore and pore structures. a** Domain organization of PFN2: SP: signal peptide, MACPF: light purple (including TMH1 and 2: transmembrane hairpin 1 and 2, red), EGF: light purple, P2: green, CTT, C-terminal tail; TM: transmembrane helix; Cyto: cytosolic tail. **b** and **f** PFN2 pre-pore (**b**, Protein Data Base (PDB) 6SB3) and pore (**f**, PDB 6SB5) structures. Domains colored as in **a**. Top: side view; middle: top view; bottom: side view slice of subunits 1 and 2 as indicated by the dashed line in the top view. **c** and **g** Pre-pore (**c**) and pore (**g**) structures in ribbon representations of subunits 1 and 3 as indicated in **b** and **f**. Comparison of pre-pore and pore indicate the 180° flipping of the MACPF domain in the conformational transition. **d**, **e** HS-AFM images of PFN2 pre-pore oligomers on mica (**d**, experimental

repeats, $n = 3$) and *E. coli* lipid membrane (**e**, experimental repeats, $n = 10$). **h**, **i** HS-AFM images of PFN2 pore oligomers on mica (**h**, experimental repeats, $n = 3$) and *E. coli* lipid membrane (**i**, experimental repeats, $n = 10$). **j** Height distribution of pre-pore (mean ± std: 8.2 nm ± 0.4 nm) and pore (mean ± std: 14.0 nm ± 0.5 nm) oligomers on mica. **k** Assembly size distribution of pre-pore and pore oligomers on mica. **l** Height distribution of pre-pore (mean ± std: 8.1 nm ± 0.2 nm) and pore (mean ± std: 12.1 nm ± 0.6 nm) above *E. coli* lipid membrane. **m** Assembly size distribution of pre-pore and pore oligomers on lipid membrane. The distributions in (**j**, **k**, **l**, **m**) are labeled with the assembly species analyzed and the respective data (**d**, **e**, **h**, **i**). Source data are available as a Source Data file.

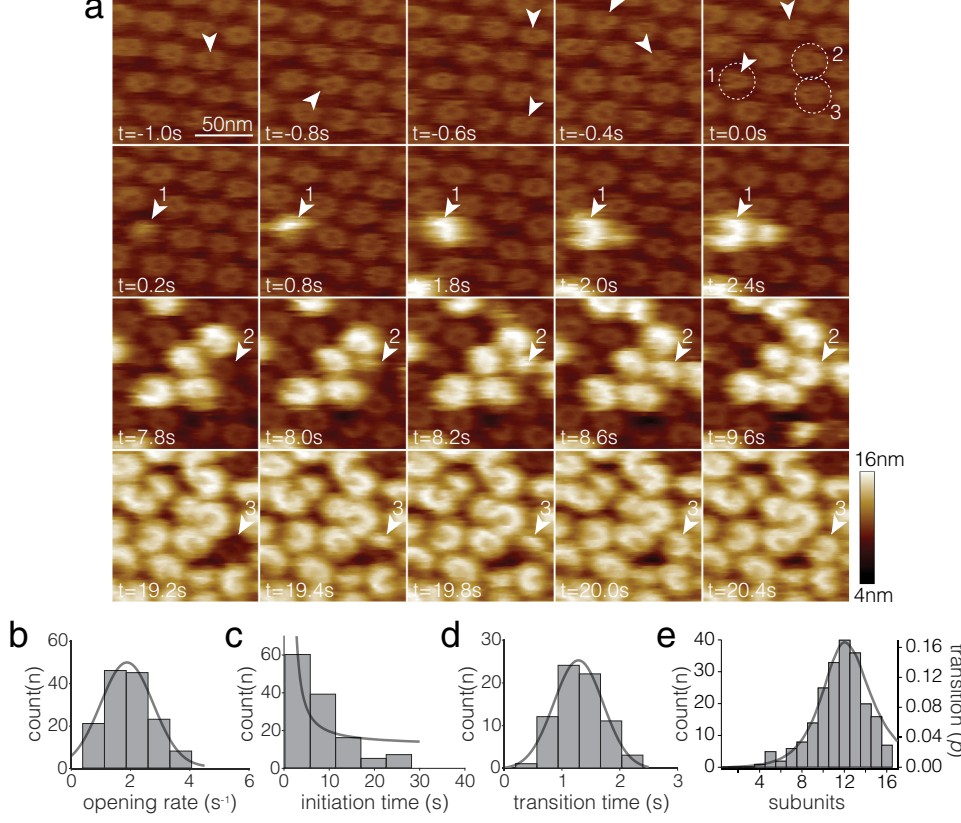

**Fig. 2 | HS-AFM imaging of PFN2 clockwise (CW) pre-pore to pore transition.**
**a** Frames from a HS-AFM movie (Supplementary Movie 2; time $t = 0$ s is set to the moment when the first subunit in the first pre-pore initiated a conformational transition) at 200 ms temporal resolution, focusing on four specific time intervals: 1st row ($t = -1.0$ s to $t = 0.0$ s): PFN2 pre-pore rings destabilized through acidification at pH 4.0. Circles emphasize the three molecules further investigated in the rows below. Arrowheads indicate pre-pore rings inter-subunit breakage sites. 2nd ($t = 0.2$ s to $t = 2.4$ s), 3rd ($t = 7.8$ s to $t = 9.6$ s) and 4th ($t = 19.2$ s to $t = 20.4$ s) rows: Pre-pore to pore transitions of assemblies 1, 2 and 3, respectively. All transitions propagate CW (experimental repeats, $n = 8$). **b** Opening rate distribution of pre-pore oligomers upon acidification, $1.8 \pm 1.2$ s$^{-1}$ (mean ± std). **c** Pre-pore survival lifetime distribution and single exponential decay fit, after pre-pore to pore transition initiation upon acidification: Lifetime constant: ~5 s. **d** Pre-pore to pore transition time distribution $1.3 \pm 0.6$ s (mean ± std). **e** Pre-pore to pore subunit transition length distribution and fit (see Methods; $n_O = 11.9$, $p = 0.64$). Source data are available as a Source Data file.

imaging frame (at 200 ms frame rate). Thus, we estimate that the closing rate of these breakages was »5 s⁻¹. At pH 7.5 such opening events almost never occurred (Supplementary Movie 1). Thus, at low pH, protonation of amino acid residues at the subunit-subunit interface (Supplementary Fig. 2), likely destabilizes the integrity of and primes pre-pore rings for the structural transition (see below for further details).

A few seconds after acidification, one PFN2 pre-pore subunit next to a ring-opening site (arrowhead and dashed outline 1 in $t = 0.0$ s), displayed a height increase ($t = 0.2$ s) and induced the transformation of the pre-pore ring into a pore arc. The initiation of this first pre-pore to pore transition was set as time zero. Plotting the time for each pre-pore to start their pre-pore to pore transition, gave a transformation initiation time constant of ~5 s (Fig. 2c). The detailed transition of three

assemblies is highlighted (Fig. 2a, arrowheads 1, 2 and 3): The pre-pore to pore transition was completed within 2.2 s ($t = 0.2$ s to $t = 2.4$ s) for assembly 1, 1.8 s ($t = 7.8$ s to $t = 9.6$ s) for assembly 2, and 1.2 s ($t = 19.2$ s to $t = 20.4$ s) for assembly 3. In general, the transition time was thus quite consistent among assemblies, averaging at ~1.3 s (Fig. 2d). To interpret the pore arc-length distribution, we elaborated a simple model that fits the pore transition run length distribution, based on a coupling probability, a value that characterizes the likelihood of subunit $n + 1$ to transition into the pore conformation following the pore transition of subunit $n$, and a penalty for long runs leading to closed pore rings (Fig. 2e, see Methods). Based on this model, we estimated that each CW-positioned subunit had a probability of ~0.64 to transit into the pore state after the neighbor subunit transitioned into the pore state. The fit peaks at 11.9 subunits, in good agreement with the

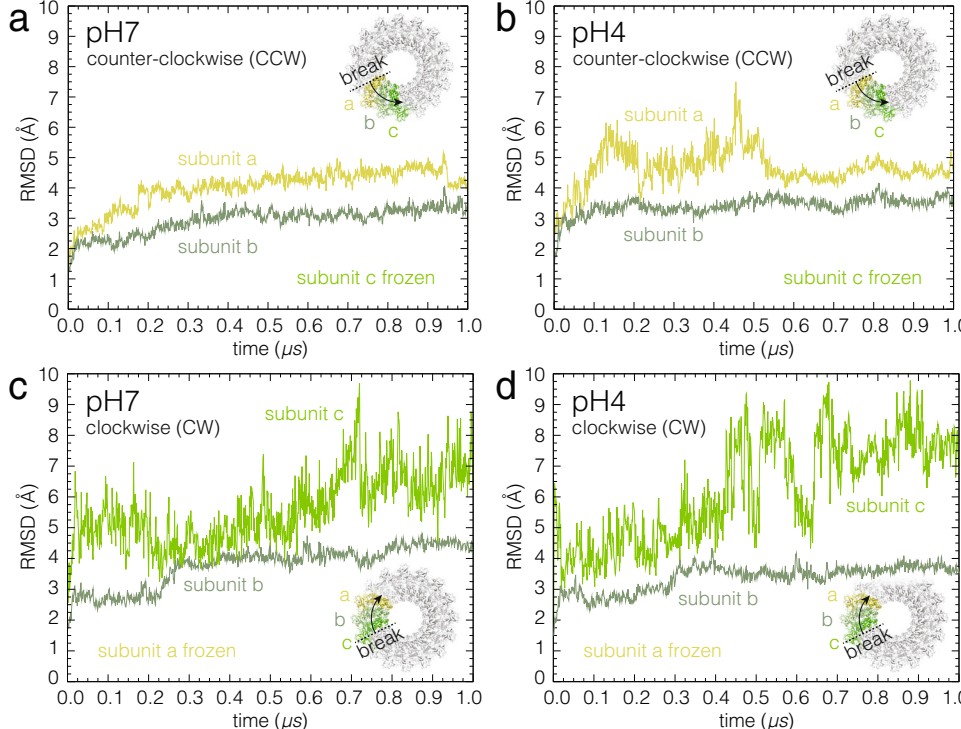

**Fig. 3 | Structural fluctuations of the free subunits upon pre-pore ring breakage.** Comparison of the backbone atom distance root-mean-square deviations (RMSDs) of the CCW (subunit-a, **a**, **b**) and the CW (subunit-c, **c**, **d**) subunits with respect to their initial structures at neutral (**a**, **c**) and acidic (**b**, **d**) pH. Source data are available as a Source Data file.

mode value, 12, of the distribution. For reasons that need further investigation, very long runs, >12 subunits, are rare (see Discussion), and only few pores formed complete rings (Supplementary Fig. 3a). However, ring breakage could spontaneously occur at two or more sites within a pre-pore oligomer. And indeed, we found rare experimental evidence for this to happen; where two breakages initiated pre-pore to pore transitions separately, which ultimately reunited into a larger pore arc (Supplementary Fig. 3b).

When pre-pore fluctuations could be observed (molecule 1), the transition started from the breakage site and propagated in a CW subunit sequence, viewed from above the membrane (Fig. 2a, molecules 1, 2 and 3, Supplementary Movie 2). The CW propagation of the pre-pore to pore transformation means on the structural level that the concave subunit surface transfers the conformational transition to the convex surface (Fig. 1c, Supplementary Fig. 2b, bottom) and not in the opposite direction (Supplementary Fig. 2a, bottom). PFN2 is thus a unique system that allows direct interrogation of the structure and dynamics of the end-states as well as the transition process itself.

### The clockwise free subunit is unstable and pH sensitive

To assess why the PFN2 pre-pore to pore conformational transition occurred CW and not in a CCW direction, we investigated the impact of acidification on the subunits on either side of a pre-pore ring breakage using molecular dynamics simulations (MDS). For computational feasibility, we monitored the structural fluctuations of the subunits in a pre-pore trimer (Fig. 3, subunits a,b,c), where either subunit-c or subunit-a are held in space to mimic the continuation of the pre-pore ring to either side. We reasoned that the greater fluctuations in an end positioned, untethered subunit mirrored a stronger perturbation, and, hence, a higher likelihood to transition from pre-pore to pore. As viewed from above the lipid bilayer, the trimer with either subunit-a or subunit-c spatially tethered corresponds, respectively, to a CCW (Fig. 3a, b) or CW (Fig. 3c, d) propagation of the perturbation.

Structural fluctuations are commonly measured as the root-mean-square deviation (RMSD) with respect to a reference conformation, from a suitably thermalized computational assay. Fluctuations of subunit-c were more pronounced when propagation was CW (Fig. 3c, d), compared to subunit-a fluctuations in a CCW setting (Fig. 3a, b), irrespective of the pH. Lowering the pH further amplified the fluctuations in subunit-c (Fig. 3d). To identify where in the free subunit these fluctuations were primarily located, we analyzed the RMSDs of the different protein domains (Supplementary Fig. 4). This analysis showed that the MACPF, TMH1, TMH2 and EGF domains formed a robust core, with low RMSDs and rather insensitive to the change of pH, independent of the CCW or CW setting. In both CCW or CW perturbations, the CTT domain was strongly destabilized upon acidification. This result is noteworthy, because the CTT (i) was close to the membrane in the pre-pore state, and (ii) was not resolved in the pore-state structure[44, 45]. However, most importantly, with a CW perturbation, the largest fluctuations were observed in the P2 domain, that was stable in the CCW setting. In other words, the low pH-dependent dynamics of the two solvent-accessible surfaces freed by pre-pore ring breakage varied markedly: The free surface of the CW subunit is highly mobile, requiring a partner subunit to stabilize it, in stark contrast to the free CCW subunit, which is intrinsically stable. While the MACPF domain undergoes later during the conformational transition the largest conformational changes, during these initial steps of the process, accessible to MDS, we found the major fluctuations in other domains, notably the domains, CTT and P2, that are in proximity to the membrane in the pre-pore state.

### Inter-subunit salt-bridges are pH sensitive

These first MDS provided a basis for the experimental finding that upon acidification, inducing intermittent inter-subunit breakages, the pre-pore to pore transition originated at the CW located subunit, because it was intrinsically unstable. But what is the mechanism by which the pre-pore to pore transition is propagated between subunits?

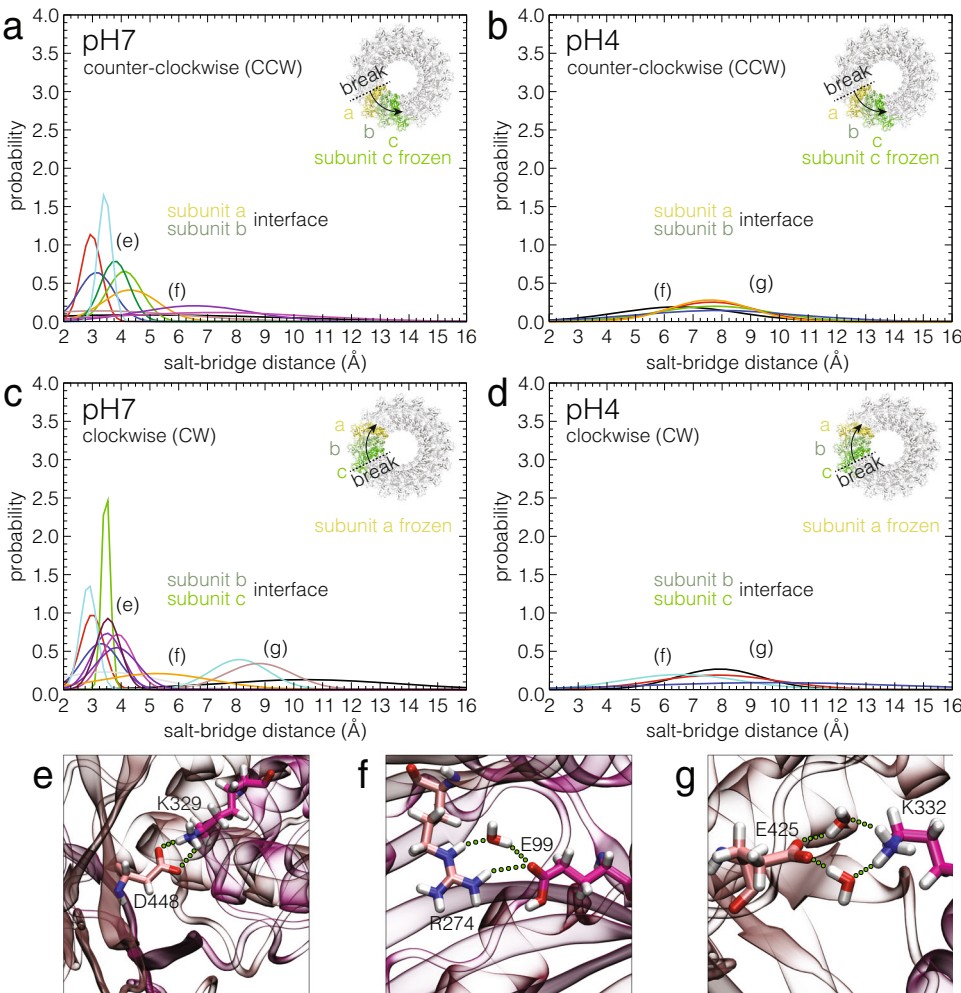

**Fig. 4 | Inter-subunit salt bridges are pH sensitive.** Comparison of the probabilities of occurrence of interfacial salt bridges at neutral (**a, c**) and acidic (**b, d**) pH inferred from μs-timescale molecular dynamics trajectories, between the first and second subunit in CCW (subunit a-b interface, subunit c frozen) or in CW (subunit b-c interface, subunit a frozen) direction. Each line represents an individual salt bridge. The salt bridges are labeled with (**e, f, g**) according to their carboxylate separation distances as shown in the panels below, (**e**) -3.5 Å, (**f**) -6 Å, (**g**) -8 Å. Each line is normalized: $\int dx\, h(x) = 1$. Direct (**e**) and water-mediated (**f, g**) salt bridges are identified based on the distance separating the carboxylate oxygen atoms of aspartate (D) and glutamate (E) from the nitrogen atoms of lysine (K) and arginine (R)[76]. Source data are available as a Source Data file.

To address this question, we mapped the electrostatic potential of titratable amino acids at the inter-subunit interface in response to a pH-decrease from pH 7 to pH 4. At pH 7, irrespective of subunit-c (CCW) or subunit-a (CW) being positionally fixed, the contiguous interacting surfaces a-b (CCW) or c-b (CW) exhibit a near ideal complementarity between negative and positive electrostatic-potential regions (Supplementary Fig. 2). Notably, multiple long-lived salt bridges, most direct and some mediated by water molecules, formed between these subunits (Fig. 4a, c, Supplementary Table 1). Conversely, at pH 4.0, all direct salt bridges disappeared, and the number of water-mediated interactions was perceptibly reduced (Fig. 4b, d), consistent with the overall positive electrostatic potential surfaces repulsing each other (Supplementary Fig. 2, Supplementary Table 1). This result emphasizes the critical role of the protonation of titratable amino acids at the subunit interfaces as well as the net weakening of electrostatic attraction between subunits at low pH. Comparing the mean occupancies of salt bridges (Supplementary Table 1), lowering the pH results in a greater variation when the perturbation progresses CW ($\Delta = 61$ direct and $\Delta = 5$ water-mediated interactions), than with a perturbation progressing CCW ($\Delta = 45$ direct and $\Delta = 9$ water-mediated interactions). Thus, the CW interaction is weakened, and as a result, the next CW-positioned subunit has an enhanced propensity to undergo

the pre-pore to pore transition (Fig. 3), and so on, leading to a coordinated hand-over-hand CW directional transition mechanism.

## Three-step PFN2 pre-pore to pore transition

To gain further insights into the pre-pore to pore transition, we analyzed single PFN2 assemblies in the HS-AFM data, and found that PFN2 oligomers proceeded in a three-step transition process from pre-pore to pore state (Fig. 5a–c): First, acidification triggers pre-pore rings to break open to form an arc of 16 subunits (or two shorter arcs). The specific location of the breakage site along the ring appeared to be random with no discernable pattern. Second, next to a breakage site the pre-pore subunit undergoes a structural transformation reported in HS-AFM by a height increase of ~1.6 nm. We termed this state pre-pore-II, while we termed the original pre-pore state pre-pore-I. As briefly discussed above, the transition from pre-pore to pore state necessitates two large conformational transitions: (i) a 180° flipping of the MACPF domain (according to the cis-model[44, 45]), and (ii) unfurling of the TMHs into β hairpins. These transitions occur in this order, because flipping of the MACPF domain must first bring the TMHs to the membrane surface. Therefore, we propose that the transition from pre-pore-I to pre-pore-II corresponds to the MACPF domain flip (Fig. 5c). Once one subunit adopts the pre-pore-II state, coupling to the

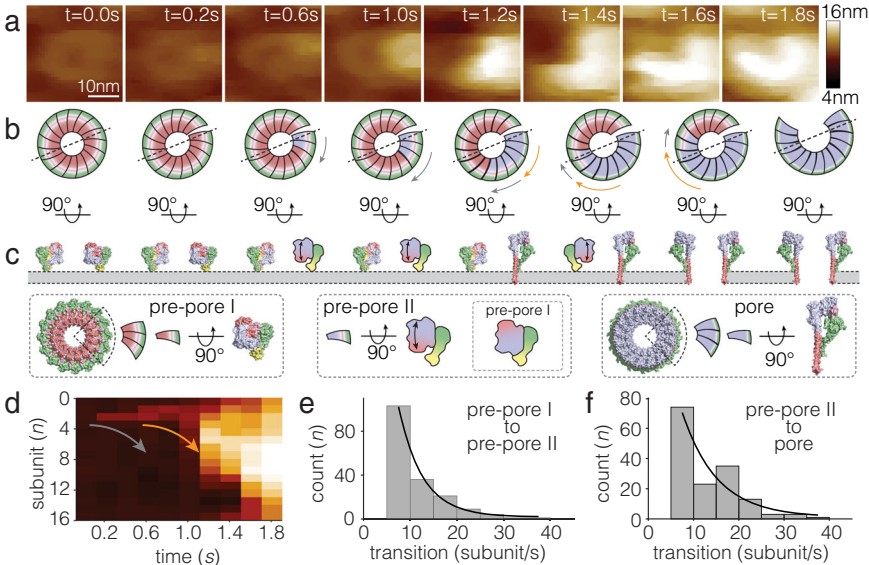

**Fig. 5 | Three-step process of PFN2 pre-pore to pore transition. a** In situ HS-AFM frames of the pre-pore to pore transition process. **b** Top view schematics and **c** side view structures and schematics of the pre-pore to pore transition in **a**. In the side view (**c**) pre-pore-I and pore states are represented by the corresponding structures, while pre-pore-II is represented by a schematic of a pre-pore with a flipped MACPF domain. Bottom: explanation of the pre-pore-I, pre-pore-II and pore state structures and schematics. **d** 2D-image of clockwise (CW) pre-pore to pore transition. Arrows roughly illustrate the CW transition of pre-pore I to pre-pore II and pre-pore II to pore. **e, f** Histogram analyses and exponential decay fits of PFN2 transition rates of pre-pore-I to pre-pore-II, ~13 s$^{-1}$, and pre-pore-II to pore, ~15 s$^{-1}$.

CW adjacent subunit induces pre-pore-I to pre-pore-II transition in this subunit (Fig. 5a, b). While we do not have a pre-pore-II structure, it appears reasonable to hypothesize that flipping of the MACPF domain in one subunit inverts the surface that it exposes to its CW neighbor, and thus flipping of the MACPF domain in the following subunit may allow reformation of some of the former inter-subunit wedging and contacts, though these contacts should still be weak as the surfaces are protonated (see Supplementary Fig. 2, bottom). Thus, we imagine the propagation as a chain-reaction of inter-subunit interface breakages that favor the transition of the next exposed subunit. Third, following the pre-pore-I to pre-pore-II transition, pre-pore-II undergoes further structural transformations to adopt the final pore state. The pore state is characterized by a further height increase by ~2.2 nm compared to pre-pore-II. We infer that the transition of pre-pore-II to pore should implicate TMHs unfurling into the long extended β-hairpins (independent of the model of interpretation) (Fig. 5c). The overall ~3.8 nm height increase of the pore compared to the pre-pore-I state is in excellent agreement with the total height difference between the pre-pore and the pore structures (see Fig. 1b, f). The CW transition from pre-pore-1 to pre-pore-II and to pore can be visualized by computationally straightening the assembly into a column of 16 pixels and plotting the binned average height in each pixel as a function of time (Fig. 5d). The subunit transition rate from pre-pore-I to pre-pore-II was ~13 s$^{-1}$ (Fig. 5e), while the transition from pre-pore-II to pore was estimated to ~15 s$^{-1}$ (Fig. 5f). Thus, the slightly slower pre-pore-I to pre-pore-II transition propagates ahead of the pre-pore-II to pore transition. The high transition rates of both processes, pre-pore-I to pre-pore-II and pre-pore-II to pore, imply a complete transition of the PFN2 oligomer in ~1.3 s, consistent with the HS-AFM imaging observations (see Fig. 2d).

**Millisecond temporal resolution of PFN2 pre-pore to pore transitions**

To further investigate the transition dynamics of PFN2, we performed HS-AFM line scanning (HS-AFM-LS)[55] experiments at room temperature with 2 ms temporal resolution during PFN2 pre-pore to pore transition (Fig. 6a): In a line-scanning experiment, instead of recording one frame with 500 scan lines per second, we recorded the same scan line 500 times per second and arranged the scan-lines as a function of time to produce a topography kymograph. First, an image was taken where all PFN2 assemblies were hexagonally packed rings in the pre-pore state (Fig. 6a, left). Second, upon acidification of the buffer initiating of the transition process, the y-scan axis was disabled and the height profile of the PFN2 subunits in the central region of the former image was recorded at high temporal resolution to provide a time course of the protomer height evolution (Fig. 6a, center). Returning into imaging mode allowed to confirm that all PFN2 oligomers had transited into pore state during the HS-AFM-LS experiment (Fig. 6a, right). At the increased temporal resolution of HS-AFM-LS all states, pre-pore-I, pre-pore-II and pore, could be resolved with discernable height plateaus (Fig. 6b). The height increments from pre-pore-I to pre-pore-II was 1.6 ± 0.4 nm (Fig. 6c) and from pre-pore-II to pore was 2.2 ± 0.2 nm (Fig. 6f), consistent with the afore-mentioned HS-AFM imaging data. This experiment also allowed us to get an estimate of the speed of the conformational change with raising speeds from pre-pore-I to pre-pore-II of 47 ± 5 nm/s (Fig. 6d) and from pre-pore-II to pore of 48 ± 7 nm/s (Fig. 6g), respectively. Considering that the amino acid length pitch in a β-strand is 1.75 Å, and the bilayer is ~40 Å thick (corresponding to a transmembrane β-strand length of ~23 amino acids), we estimated that the unraveling of the TMHs into β-strands during the pre-pore-II to pore transition of ~46 ms occurred at a velocity of ~0.5 amino acids per millisecond. The lifetime of the pre-pore-II state was fit with an exponential decay with 75 ms time constant (Fig. 6e). Based on its short lifetime, we think that pre-pore-II is a high-energy transition state, and that the unfurling of TMHs into the extended membrane-embedded β-hairpins for pore-formation is associated with an energy gain.

## Discussion

HS-AFM has proven powerful for characterizing protein dynamics by providing unique direct real-time dynamic and structural information. Typically, HS-AFM has allowed characterization of the kinetics of the interconversion of major conformational states. Here, we report two unique achievements using HS-AFM. First, we could characterize an intermediate state, pre-pore-II, which has a lifetime of only ~75 ms. We think it will be challenging for other structural techniques to capture

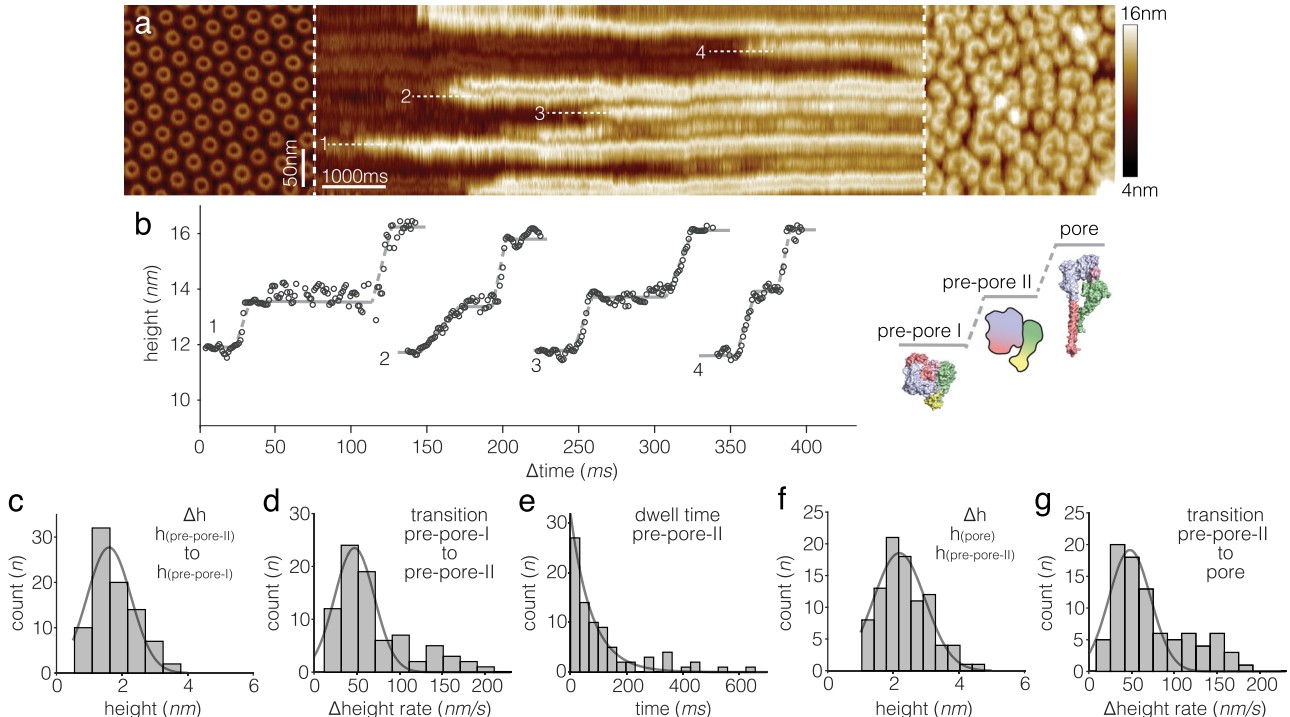

**Fig. 6 | HS-AFM line scanning (HS-AFM-LS) with millisecond temporal resolution of PFN2 subunit pre-pore to pore transition.** **a** Left: HS-AFM image of PFN2 pre-pore oligomers (rings) before HS-AFM-LS. Center: HS-AFM-LS of PFN2 pre-pore-to-pore transition (line acquisition rate: 500 lines/s, 2 ms/line temporal resolution). Right: HS-AFM image of PFN2 pore oligomers (arcs) after HS-AFM-LS (experimental repeats, n = 20). **b** Height profiles 1–4 of subunit pre-pore-I to pre-pore-II to pore transitions along the dashed lines 1–4 in **a**. Inset: cartoon of PFN2 pre-pore-I to pre-pore-II to pore transition (schematic as in Fig. 5). **c**, **f** Histograms of pre-pore-I to pre-pore II (**c**, mean ± std: 1.6 ± 0.4 nm) and pre-pore-II to pore (**f**, mean ± std: 2.2 ± 0.2 nm) subunit height increases. **d**, **g** Histograms of pre-pore-I to pre-pore-II (**d**, mean ± std: 47.3 ± 5 nm/s) and pre-pore-II to pore (**g**, mean ± std: 48 ± 7 nm/s) height increase speeds. **e** Lifetime distribution of the pre-pore-II state and exponential fit with a lifetime constant of 75 ms. Source data are available as a Source Data file.

and characterize such short-lived intermediates in native protein structures. Second, we discovered a stereotyped CW structural transition. This is distinct from imaging two end-states, as it directly indicates the structural path connecting the two major states, pre-pore and pore. By identifying a specific and physically meaningful transition pathway out of a vast number of possibilities, the findings presented based on our HS-AFM experiments should be helpful for structural modeling of the interconversion of PFN2 states. It is noteworthy, that the structures of MAC, which is a hetero-oligomeric complex, are, based on the arrangement of the subunits along the pore, evocative of a CW assembly[41, 46].

MDS revealed that at acidic pH, due to protonation of titratable residues, inter-subunit salt bridges were undone, leading to destabilization of and breakages in the pre-pore ring. Interestingly, subunits on either side of a breakage are not equally destabilized: Fluctuations in the distance RMSD of backbone atoms are about twice as large in the subunit located CW from a breakage, as compared to a subunit located CCW. This asymmetry primes the CW subunit to initiate the pre-pore to pore transition. In addition, the greater perturbation of inter-subunit salt bridges in the CW direction primes the next CW subunit for pre-pore to pore transition. In more detail, particularly strong fluctuations were detected in the CW subunit P2 and CTT domains that anchor the pre-pore ring to the membrane. Thus, the structural fluctuations and the breakage of inter-subunit salt bridges both amplified in the subunit located CW from a pre-pore ring opening are likely at the basis of the specific CW PFN2 pre-pore to pore transition.

Together, our HS-AFM imaging and HS-AFM-LS experiments give the following picture of the PFN2 pre-pore to pore transform (Fig. 7, Supplementary Table 2). Upon activation through exposure to low pH, the inter-subunit contacts in PFN2 pre-pore rings are destabilized and occasionally break at a rate of ~1.8 s$^{-1}$, into an arc of all 16 subunits (or

rarely shorter arcs, when multiple breakages occurred). Closing these breakages without a major consequence to the pre-pore assembly must occur frequently, at least at a rate of >5 s$^{-1}$. However, some pre-pore assembly breakages serve to initiate the pre-pore to pore transition: Starting from the subunit located CW from the breakage site, the pre-pore subunit transitions from the pre-pore-I to the pre-pore-II state. This happens upon acidification roughly with a rate of ~0.2 s$^{-1}$ in each pre-pore ring. Because pre-pore rings are very stable at neutral pH and rather slow to initiate a transition at acidic pH, ~0.2 s$^{-1}$, we propose that pre-pore assembly/oligomerization are separated from the pre-pore to pore transition, which is fast. This initial conformational change to a pre-pore-II subunit primes the CW adjacent subunit to undergo the same conformational change. The CW propagation of pre-pore-II is fast, at a rate of ~13 s$^{-1}$. The conformational state of pre-pore-I to pre-pore-II is characterized by a height increase of ~1.6 nm at a speed of ~47 nm/s. We do not have a structure of pre-pore-II, but hypothesize that the ~1.6 nm height increase is the manifestation of the MACPF flipping, which would accordingly be achieved within ~34 ms time. However, as the pre-pore-I to pre-pore-II transition propagates CW, the very first pre-pore-II subunit starts to transform to the final pore state. This observation explains why the pre-pore-II state is only an intermittent state with a short lifetime of ~75 ms. The pre-pore-II to pore subunit transition is equally CW coupled and propagates swiftly to the adjacent subunits at a rate of ~15 s$^{-1}$, until all the subunits achieve the final pore structure. The conformational change from pre-pore-II to pore is manifested by a height increase of ~2.2 nm at a speed of ~48 nm/s, which most likely corresponds to the unfurling of the extended TMH1/TMH2 β-sheet and piercing of the membrane, which accordingly occurs within ~46 ms time, at ~0.5 amino acids per millisecond. Because the final pores are not entire rings, but rather arcs with varying length peaking at 12 subunits, we estimated the pre-pore to pore

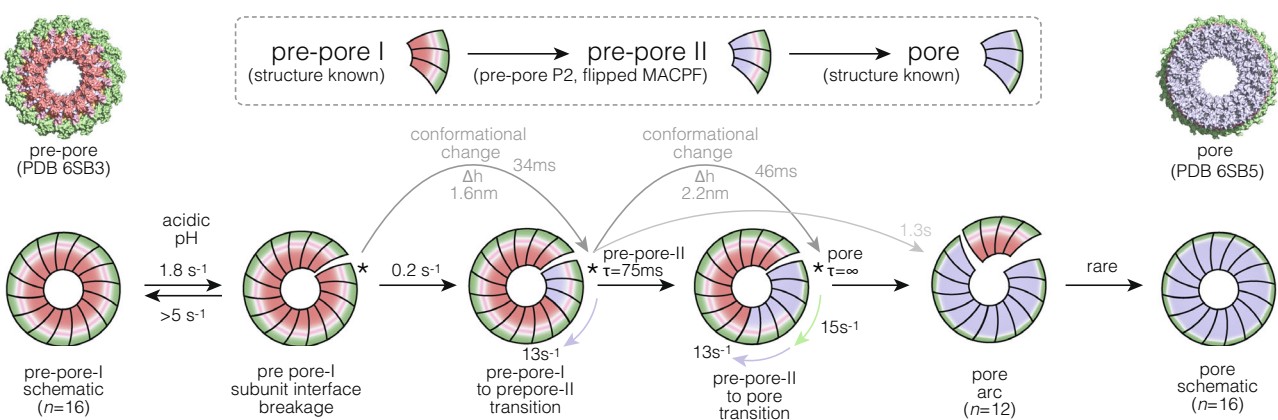

**Fig. 7 | PFN2 pre-pore to pore transition mechanism and kinetics.** HS-AFM imaging with 200 ms temporal resolution and HS-AFM line scanning (HS-AFM-LS) with 2 ms temporal resolution allowed the detailed characterization of the PFN2 pre-pore to pore transition mechanism.

transition coupling to be -0.64. We do not know why longer pre-pore to pore transition runs are rare, but the pore length distribution could be well modeled with the addition of an energy penalty for long runs. We hypothesize that pore arcs of -12 subunits, three-quarters of the entire ring, might readily lead to the formation of an aqueous pore in the membrane, as is observed for other arc forming PFPs[24] and, that, as a consequence, the energy gain of the pre-pore to pore transition of the latter subunits, >12, might be diminished. Put together, except for the initial step, the occasional openings/breakages within pre-pore rings, all steps were irreversible, and pores exposed to physiological pH never reverted to pre-pores.

In a cellular setting, where the local PFN2 subunit concentration might be limiting, pre-pore formation might stop at the stage of an arc and breakage may not be needed to initiate the pre-pore to pore transition. But the pH-induced increased destabilization of the CW exposed subunit and the loosening of the inter-subunit packing would still favor the CW initiation and propagation of the transition.

A structural comparison of PFN2 with other MACPF/CDC pore forming proteins showed high structural similarity (Supplementary Fig. 5)[43,56–61], especially regarding the MACPF domain which is involved in pore formation and membrane penetration. The central, common feature of the MACPF/CDC fold is a four-stranded, twisted β-sheet and three clusters of α-helices. Two of these helical bundles contain the regions destined to insert into the membrane (TMH1 and TMH2) to generate the four-stranded β-sheet of each subunit that forms with the same elements from the other subunits the large transmembrane β-barrel (Supplementary Fig. 5). These similarities of key structural elements, in particular the architecture of the final pore state[44], together with the common ancestor and the shared pore-forming function, suggest that all MACPF/CDC proteins might share features of the transition mechanism unraveled here: Indeed, although PFN2 is unique among the described MACPF/CDC proteins in that its membrane-binding and pore-forming domains face in opposite directions in the pre-pore oligomer, all members of the family must refold from a globular fold into a β-barrel structure, suggesting that breaks between subunits in the pre-pore ring, and CW propagation of membrane insertion might be common features.

## Methods

### Protein expression and purification
All the proteins in this study were produced recombinantly from mammalian HEK293T or HEK293S cells using a transient transfection and expression protocol as described[62,63]. The proteins were flash-frozen in liquid nitrogen and stored in −80 °C until further use. Ecto-domain of mPFN2 (Uniprot: A1L314, residues 20–652) was purified

from the media using anti-1D4 antibody-coupled agarose resin as described previously[44].

### Formation of supported lipid bilayers (SLBs) for HS-AFM
The SLB formation method was detailed in our previous study[64]. In brief, first lipid vesicles were prepared from 100% *E. coli* total phospholipid extract, composed of PE: PG: CA: unknown = 57.5: 15.1: 9.8: 17.6 (w/w; Avanti, USA). Lipids dissolved in chloroform were dried under argon flux, followed by >2h incubation in a vacuum desiccator. Following, lipids were fully rehydrated with buffer (10 mM Hepes-NaOH, pH 7.5, 200 mM NaCl) for 5 min at room temperature, with adjusted volume to give a lipid solution at 0.5 mg/ml. Next, the lipid suspension was bath-sonicated for 10 min to form vesicles. Finally, to form SLBs on mica, 3 μl of the *E. coli* lipid solution was deposited onto a freshly cleaved 1.5 mm diameter mica disc, incubated for 15 min, and rinsed thoroughly with buffer (10 mM Hepes-NaOH, pH 7.5, 200 mM NaCl, 2 mM CaCl₂).

### High-speed atomic force microscopy (HS-AFM)
HS-AFM imaging of PFN2 assemblies on mica or on lipid bilayers was performed at room temperature by incubating PFN2 monomers (0.043 mg/ml) for 5 min, either directly on mica or on preformed *E. coli* lipid SLBs. Then, the sample was rinsed and directly imaged by HS-AFM. All images in this study were acquired using a HS-AFM (SS-NEX, RIBM, Japan) operated in amplitude modulation mode using a home built fast amplitude detector[65] and a home built free amplitude stabilizer force controller[66] and optimized scan and feedback parameters. 8 μm short cantilevers (USC-F1.2-k0.15, NanoWorld, Switzerland) with nominal spring constant of 0.15 N/m, resonance frequency of -0.6 MHz in buffer, and a quality factor of -1.5 in buffer (20 mM Hepes-NaOH, pH 7.5, 200 mM NaCl, 2 mM CaCl₂) were used. To study the pre-pore to pore transition, the fluid cell containing -120 μl of buffer (20 mM Hepes-NaOH, pH 7.5, 200 mM NaCl, 2 mM CaCl₂) was supplemented with -15 μl of 0.1 M HCl (through passive diffusion) to decrease the pH of the buffer solution to -pH 4.0. The change in pH was estimated by large-scale mixing 150 μl of 0.1 M HCl into 1.20 ml of the imaging buffer in parallel to the HS-AFM experiments. HS-AFM movies were plane fit flattened in ImageJ 1.52.

### HS-AFM line scanning (HS-AFM-LS)
HS-AFM-LS was performed at room temperature upon disabling the slow scan axis (*y*-axis) of scanning[55,67]. In brief, after taking an image at a rate of 1 frame per second with 500 scan lines (Fig. 6a, left), the *y*-piezo scan range is set to 0 while maintaining the *y*-piezo offset voltage to the same value as in the precedent image. Consequently, the central *y*-scan line is repeatedly scanned at a rate of 500 times

per second, and stacked as a function of acquisition time, resulting in a section profile kymograph with 2 ms temporal resolution (Fig. 6a, middle). After HS-AFM-LS kymograph recording the y-scan axis is re-activated to acquire, as a control, another image of the scan area (Fig. 6a, right).

## Determination of kinetic parameters

**Pre-pore-I inter-subunit breakages.** In HS-AFM movie frames recorded at 200 ms temporal resolution and at acidic conditions (e.g. Fig. 2a, frames $t = -1.0$ s to $t = 0.0$ s), we counted the number of pre-pore rings with a visual discontinuity of the ring structure and divided this number by the total number of ring observations during time windows of 1 s. We found openings in average at a rate of $1.8$ $s^{-1}$ (Fig. 2b). We never observed openings to persist over more than a single frame and thus, we estimate the lowest bound of the reformation rate as $5$ $s^{-1}$ (each scan line is acquired at 1 ms temporal resolution).

**Pre-pore-I activation.** In HS-AFM movie frames recorded at 200 ms temporal resolution and at acidic conditions (e.g. Fig. 2a, frames $t = 0.0$ s to $t = 20.4$ s), we counted the number of pre-pore rings before they transited into the pore state, setting $t = 0.0$ s from the moment the first transition dynamics in the first ring was observed. The distribution is shown in Fig. 2c and a single exponential fit for the decay of the pre-pore state gave a characteristic lifetime of 5 s that translates, considering a first order reaction, to a rate of $0.2$ $s^{-1}$. Thus, the nucleation probability to initiate a pre-pore to pore transition is actually rather low, as we estimate inter-subunit breakages to occur at a rate of $-1.8$ $s^{-1}$ in each ring (see above), but the transition only occurs a rate of $0.2$ $s^{-1}$.

**Coupling of pre-pore to pore transition.** From the pore arc-length distribution (Fig. 2e), we estimated a coupling function for subunit $n + 1$ to transits to the pore state after subunit $n$ transited to pore. The number of subunits in the pore arcs in Fig. 2e were calculated by fitting the arc length of the pore structures, where a full ring of 360° would correspond to 16 subunits and an arc of e.g. 270° to 12 subunits. Initially, all $N = 16$ subunits are in the pre-pore state and could undergo a transition to the pore state. For the first subunit, the probability ($\sigma$) to transit from pre-pore to pore is low ($\sigma \ll 1$; it happens only rarely and most often only once in a ring), and thus the energy barrier it has to overcome in the process must be high. However, once subunit-($n$) has made this transition, then the CW neighboring subunit-($n + 1$) can undergo the pre-pore → pore transition with a probability proportional to the Boltzmann factor $s = exp(-\Delta E/kT)$, where $\Delta E$ is the equilibrium energy difference between the pore and the pre-pore states in this setting. We assume $\Delta E < 0$, so that the Boltzmann factor $s > 1$ (the probability for subunit-($n + 1$) to transition is favorable after subunit-($n$) transited into the pore state). This treatment is akin to the Zimm-Bragg model of helix nucleation and propagation in a random coil peptide[68]. Thus, for a run to occur and propagate for $n$ consecutive subunits to transitioning into the pore state, we would get a probability proportional to $\sigma \cdot s^n$. However, two additional factors should also be considered. First, before the transition process, all subunits were equal, and therefore we consider them indistinguishable. As a consequence, there were 16 ways for runs of any length $n = 1, 2, ..., 16$ to occur. Second, we might need to consider another energy term as the run of consecutive transitions advances around the 16-mer. If there was some induced energy penalty (e.g. strain in the protein assembly, membrane distortion, or membrane pore formation) that accumulates as the pre-pore → pore transition advances around the ring, then a penalty term should be included in the probability expression, so that runs of long lengths (say $>n = 12$, Fig. 2e) are less favored. A simple approach is to use another Boltzmann factor based only on the run length and some

characteristic length ($n_0$) associated with the penalty energy: $1/(1 + exp(n-n_0))$. This is a phenomenological term and not meant to model a particular source of the energy penalty. Putting together the run statistics and the geometry penalty, we generated a partition function $Z$ (Eq. (1)) for the 16-mer ring, and a probability $p(n)$ (Eq. (2)) of pre-pore to pore runs, which give a subunit transition probability $p(0)$ (Eq. (3)):

$$Z = 1 + \sum_{n=1}^{N} N \cdot \sigma \cdot s^n \cdot \frac{1}{(1 + e^{n-n_0})} \tag{1}$$

$$p(n) = \frac{N \cdot \sigma \cdot s^n}{(1 + e^{n-n_0}) \cdot Z} \tag{2}$$

$$p(0) = \frac{1}{Z} \tag{3}$$

**Overall transition time.** In HS-AFM movie frames recorded at 200 ms temporal resolution and at acidic conditions (Fig. 2a, frames $t = 0.0$ s to $t = 20.4$ s), we recorded the pre-pore to pore transition time for each assembly, from which we deduced the average transition time, $-1.3$ s (Fig. 2d).

**Pre-pore-I to pre-pore-II and pre-pore-II to pore transition rates.** In HS-AFM movie frames recorded at 200 ms temporal resolution and at acidic conditions, we analyzed the subunit transition from pre-pore-I to pre-pore-II and pre-pore-II to pore in individual PFN2 oligomers by computationally unbending the oligomers into a column of 16 pixels (Fig. 5d). The states were assigned using height thresholds (pre-pore-I to pre-pore-II: $\Delta$height $= 1.6$ nm, threshold: 0.8 nm; and pre-pore-II to pore: $\Delta$height $= 2.2$ nm, threshold: 1.1 nm (2.7 nm from pre-pore-I base level)), and transition rate distributions (Fig. 5e, f) were fitted with single exponentials giving the rates of pre-pore-I to pre-pore-II, $-13$ $s^{-1}$, and of pre-pore-II to pore, $-15$ $s^{-1}$.

**HS-AFM line scanning (HS-AFM-LS) to investigate PFN2 transition dynamics.** In HS-AFM-LS at 2 ms temporal resolution and at acidic conditions (Fig. 6a), allowed us to measure height section profiles of individual PFN2 protomers as a function of time (Fig. 6b). Height increases from pre-pore-I to pore-pore-II, $-1.6$ nm (Fig. 6c), and pre-pore-II to pore, $-2.2$ nm (Fig. 6f), were determined by measuring the height differences between discernible transition steps (Fig. 6b). Height increase speeds of pre-pore-I to pore-pore-II (Fig. 6d) and pre-pore-II to pore (Fig. 6g) were determined by measuring the steepness of the slopes between these steps. The lifetime of the pre-pore-II state, $-75$ ms, was determined by measuring the durations of the intermediate step level, pre-pore II, and fitting its distribution using a single exponential (Fig. 6e).

## Molecular dynamics simulations (MDS)

**Computational assays.** All simulations were performed using a trimer of subunits-a-b-c in water containing 150 mM NaCl concentration, with either subunit-c or subunit-a spatially tethered, thus mimicking CCW or CW transition settings, in both neutral pH 7 and acidic pH 4 conditions, corresponding to two distinct computational assays, and four molecular dynamics simulations, one for each condition. At pH 7, the molecular assembly consisted of three PFN2 subunits extracted from the pre-pore ring, and hydrated by 95,406 water molecules, which represented a total of 314,863 atoms and a simulation cell of dimensions of $-12.7 \times 14.7 \times 16.6$ $nm^3$ at thermodynamic equilibrium. At pH 4, the molecular assembly consisted of three PFN2 subunits, with their relevant titratable amino acids protonated, hydrated by 95,202 water molecules, corresponding to 314,587 atoms and roughly identical cell dimensions. On the one hand, these settings were chosen for

computational feasibility; and on the other hand, because the currently available pre-pore structure, used in these simulations, was solved in solution, because we are only interested in evaluating the pre-pore fluctuations prefacing the membrane penetrating the membrane, and in getting insights why a CW transition was preferred over a CCW transition. These goals could be readily achieved from the current settings.

**Molecular dynamics simulations (MDS) trajectories.** All MDS were performed with the scalable program NAMD 2.14[69]. The all-atom, macromolecular CHARMM36 force field[70] was used to describe the PFN2 trimer and their aqueous environment. Periodic boundary conditions were applied in the three directions of Cartesian space. The r −RESPA multiple time-step algorithm[71] was employed to integrate the equations of motion with a time step of 4 and $8 \times 10^{-15}$ s, for short- and long-range interactions, respectively. Covalent chemical bonds featuring hydrogen atoms were constrained to their equilibrium length by means of the SHAKE/RATTLE[72] and SETTLE[73] algorithms. The temperature and the pressure were maintained at 300 K and 1 atm, using, respectively, Langevin dynamics and the Langevin piston method[74]. Long-range electrostatic forces were taken into account by means of the particle mesh Ewald algorithm[75]. A 12-Å cutoff was utilized to truncate both van der Waals and short-range Coulombic interactions. Each molecular dynamics simulation was $10^{-6}$ s long, and was prefaced by a suitable thermalization of $20 \times 10^{-9}$ s. Positional tethering of either subunit-c or subunit-a in the PFN2 trimer, to mimic CCW or CW propagation, respectively, was achieved by means of harmonic potentials applied to $\alpha$-carbon atoms with a force constant of 10 kcal/mol Å². Visualization and analyses of all molecular dynamics trajectories were performed with the VMD program[76].

### Reporting summary
Further information on research design is available in the Nature Research Reporting Summary linked to this article.

## Data availability
The data that support this study are available from the corresponding authors upon reasonable request. The HS-AFM movies are provided as Supplementary Movies. The source data underlying Figs. 1j–m, 2b–e, 3a–d, 4a–d, 5e, f, 6b–g, and Supplementary Fig. 4a–d are provided as a Source Data file. Source data are provided with this paper.

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

## Acknowledgements

This work was supported by grants from the National Institute of Health (NIH), National Center for Complementary and Integrative Health (NCCIH), DP1AT010874 [S.S.] and National Institute of Neurological Disorders and Stroke (NINDS), R01NS110790 [S.S.]. The Division of Structural Biology is a part of the Wellcome Centre for Human Genetics, Wellcome Trust Core (Grant Number 090532/Z/09/Z).

## Author contributions

F.J. and S.S. designed the study; F.J. performed all HS-AFM experiments; F.J., J.S.D. and S.S. analyzed the data; T.N. and X.Y. expressed and purified the protein; F.D. and C.C. performed the molecular dynamics simulations; F.J., F.D., C.C., R.G., and S.S. wrote the manuscript.

## Competing interests

The authors declare no competing interests.
