## [Peer Review File · Nature Communications]

Perforin-2 clockwise hand-over-hand pre-pore to pore transition mechanism

Editorial Note: Parts of this Peer Review File have been redacted as indicated to remove third party material where no permission to publish were obtainedReviewers' Comments:

Reviewer #1:

Remarks to the Author:

Jiao et al present a time-resolved AFM study of Perforin-2/MPEG1, which represents one of three major mammalian immune effectors from the pore-forming MACPF/CDC superfamily. They conduct a biophysical analysis of the stepwise events during acid induced pore formation and in the process resolve key new intermediates. Remarkably, Jiao et al resolve a short-lived intermediate (half-life ~75 ms) that has not previously been identified.

Further, and distinct from prior studies, their work directly supports a clockwise propagation of the MACPF conformational change. The finding provides an important clue to the conserved shared mechanism of pore formation by MACPF/CDCs, with which seemingly disparate family members can now begin to be reconciled. For example, the Membrane Attack Complex is known to propagate in a similar clockwise fashion, however compared to Perforin and MPEG1 this appeared to be an exception (until now). Similarly, their data supports a model for arc pore formation and the recruitment of additional subunits, assuming the oligomerisation rate is sufficiently fast to add subunits to one end while the other is transitioning. (In fact these points may merit further comment in the discussion section).

The article is well written, comprehensive, and mostly the data support the conclusions (with some exceptions). I believe the article is suitable for publication in Nature Communications, with broad interest from the community. I remain sceptical about the interpretation of the data in the sole context of a cis-pore model, without entertaining alternative mechanisms, e.g. proposed by Pang et al.

Major contributions.

- Direct evidence of a propagation of unfurling is a remarkable achievement and provides new insight into the conserved mechanism of MACPF/CDCs more broadly.
- The observation of a short lived prepore intermediate is very novel, suggesting a structural rearrangement must occur to drive the propagation of TMH unfurling. In the context of perforin, MAC and other family members this may be driven simply by oligomerisation.
- Comprehensive quantitative kinetics of the MPEG1/PRF2 pore formation process, allowing future computational modelling of the process.

Major concerns.

The suggested mechanism of MACPF flipping seems highly implausible and not directly supported by the evidence in this study (which mostly related to kinetics and transition features). The authors suggest ring breaking accommodates the rotation of the MACPF domain in order to re-arrange into a cis orientation, however this does not explain how the next MACPF domain can rotate given that this would still encounter steric hindrance, i.e. enormous main-chain clashes during rotation in the context of the oligomer.

Furthermore, the suggestion implies that a stable MACPF/MACPF interactions must be broken, only to reform in an identical (yet rotated) state – which then somehow become unstable driving a conformational change. Why then would they not trigger in their original orientation? The energetics of this process seem highly unlikely, with the alternative explanation being far simpler, namely that the MACPF domain merely unfurls consistent with all other MACPF/CDC family members.

While the authors rely on the observation of a cis-like pore from cryoEM studies to support this model, this reviewer is not convinced. What if this reconstruction reflects an artefact, having been assembled in the presence of detergent and without a membrane bilayer to support and stabilise the P2 domain? Indeed, this study itself supports the notation that the P2 domain becomes highly mobile upon acidification and thus could simply be disordered in the imaged material.

The proposed mechanism raises more questions than it solves. I believe these issues could be rectified by interpreting the AFM and computational data in light of other possible mechanisms as well.

Other concerns.

- Fig 1, p4: I found the height measurements of the pore to be confusing and inconsistent with other reports. Ni et al measure the full pore height (including transmembrane region) to be only 14 nm based on cryo-EM (PDB 6SB5), where the transmembrane region accounts for ~4 nm (consistent with bilayer width) leaving ~10 nm above the membrane. Together these are consistent with the measurements on mica (Fig 1j), but not the measurements on membranes which gives a taller assembly at 12 nm above membrane (Fig 1l) or 16 nm when including the membrane (Fig 1f), i.e., a difference of between 1.5 to 2 nm. I suppose a difference of 2 nm may be within experimental error? Couldn't these data also be consistent with a trans-pore as suggested by Pang et al? These would sit roughly 8 nm above membrane as a prepore and roughly 14 nm as a pore, give or take depending on the behaviour of the transmembrane region without a target bilayer to stabilise it. I note that Ni et al measure the prepore and pore height in their previous report at 4 and 8 nm respectively (fig 5) – I assume this discrepancy is due to an incorrect reference point i.e. miscalibration. In this study, how were the heights calibrated?

- Line 215, p6, Figure 2 & MovieS2: Prepore opening/breakage is not obvious to me (perhaps untrained eye). This could be made clearer, perhaps a rotational average of prepores in the split washer (open) state? It is also not clear to me how these opening/breakages were detected and quantified. Were they counted manually? This could introduce bias.

- The suggested mechanism implies arcs would be more prone to transitioning, since opening is key to the transition of prepore to pore. Given that PRF2 assemblies on mica were typically arcs, do you see a faster transition time for these assemblies relative to complete prepores? Similarly, I would predict pre-assembled prepores placed on mica would have a reduced rate of transition, since mica would restrict the opening dynamics.

- Line 225, p8: The MD analysis suggests the CTT and P2 domains are most affected by lowering the pH (this domain is also highly mobile in cryoEM, observed by both Ni et al and Pang et al), however here the authors suggest it is the MACPF domain which undergoes a 180° flip. I don't think this is supported by the simulations. The MACPF domain has approx. constant RMSD (Fig S2c,d). Have the authors tried more sophisticated analysis, such as PCA to see what domain movements capture the largest variation components of the data? Perhaps the time scales are insufficient to see early signs of flipping, but do these show salt bridge breaking?

- Line 272, p8: Generally, the MD analysis provides a nice framework to understand the pH sensitivity of PRF2. However, I feel this conclusion would be better supported by mutagenesis and activity assays e.g. red blood cell lysis or dye leakage assay. The authors should consider targeting key salt bridges with alanine substitutions to determine whether these prepores become activated at more mild pH, e.g. D448A, E99A and K332A. Specifically, if these salt bridges are indeed critical, their loss should render the protein more active in less acidic conditions or at least increase the opening rate.

- Line 281, p9, fig 4a-d: What does each line represent in the probability distribution curves? Are these individual salt bridges? What is the unit of the vertical axis, a fraction of 0 to 1 or percentage? Many values exceed a "probability of 1".

Minor issues.

- Line 82, p2: "Previous studies suggest PFN2 can form pores in a second" – the phrasing is confusing, would suggest "Previous studies suggest PFN2 can form pores on time scales of one second". This prevents confusion with second membranes, as suggested in Pang et al.

- Line 81, p2: "dynamics remain largely unknown" - The authors should mention the studies of Parsons et al (2019, Nat Commun) and McGuinness et al (2022, bioRxiv) which both address similar topics of MACPF/CDC dynamics – especially since these reports are consistent with and further augment the findings herein (with respect to arcs and clockwise propagation of insertion).

- Line 103, p3: For completeness, please in parentheses state the alternate name for the P2 domain i.e. MABP.

- Line 121, p3: Nor was it resolved in the membrane bound prepore (Pang et al). As such fig1b is inaccurate as the CTT is shown.

- Line 151, p4: Figure 1b – Ni et al did not determine a membrane bound PRF2/MPEG1 structure; PDB 6SB3 is a soluble prepore, whose orientation is inferred. The PDB 6U2W from Pang et al should at least be included.

- Line 67, p2: First instance of acronym "PRF2" in the main text, please also include the other common name, "MPEG1". Fine for the rest to be only PRF2.

- I found interchanging between terms (i.e., lifetime, half-life, transition/initiation time, time constant, etc) somewhat confusing. I realise these are standard terms, perhaps to improve interpretability it would benefit the reader to have a summary table which includes all these quantities (including uncertainties) in one place.

- Please include height scale colour bars in all figures e.g. figs 1d, e, h, I; 2a; 5a; 6a.

- Many quantities (particularly in discussion and methods) are given without a space between the number value and unit, e.g. line 392, p12: "1.8s-1".

- Line 193, p5: "quite conserved" this term seems more appropriate for evolutionary processes, perhaps "quite consistent" would be clearer.

- Temperature is not stated in the methods or main text, these kinetics will be temperature dependent and as such it should be stated alongside the kinetic rate parameters determined.

- Perhaps this is beyond the scope of the study, but the authors should consider performing similar analyses using the disulphide trapped proteins reported in Ni et al. For example, is the prepore II intermediate still observed with PRF2K251C/G286C? Something to think about.

- Line 246-248, p7: "This result is noteworthy, because the CTT makes membrane contact in the prepore state ..." – this is inaccurate, the CTT is not resolved in the membrane bound prepore reported by Pang et al owing to conformational changes of the P2 domain β -hairpin upon membrane binding.

- Line 582, p17: "perforin" should be "perforin-2"

- Line 283, p9: "and acidic (b,d) pH" extra parentheses, should be "and acidic (b,d) pH".

Congratulations on a beautiful study,
Charles Bayly-Jones

Reviewer #2:

Remarks to the Author:

Jiao et al. have successfully captured the pH-dependent structural transition from the pre-pore to the pore state and its intermediates in the membrane pore-forming protein Perforin2 using the latest high-speed AFM techniques. Based on the HS-AFM imaging, they revealed counterclockwise conformational transition along the ring-shaped oligomer and also quantified the kinetics of the structural intermediates. Further the MD simulations have also provided a molecular structural understanding of the clockwise unidirectional conformational transition, which is quite convincing as a mechanistic explanation of the experimental facts. These results effectively demonstrate the potential of HS-AFM technique and contribute greatly to our understanding of the mechanism of pore formation in membrane pore toxins. Thus I consider that this work is worthy of publication in Nature Communications. On the other hand, I felt that the manuscript was not well organized and the explanations of some figures were quite difficult to understand, which made me stressful. I will make some comments below, which I think would make the manuscript better if revised.

1. First of all, "pre-pore" and "pore" in Fig. 1 are very confusing. I think the authors refer the structures of PFN2 observed at the different pH conditions of the solution as "pre-pore" and "pore", which made me really confusing at first. Also, because of the layout of the figures, I mistakenly thought that Figs. j and k were the results of the analysis of Figs. d and e. It would be better if the definitions of "pre-pore" and "pore" is clarified at the beginning of the manuscript, and the layout of the figures were also reconsidered.

1. I understand that the difference in brightness in the AFM image in Fig. d-i is due to the difference in height, but it would be easier to understand if a color bar could be added next to each image.

2. Several analyses account for the number of subunits of oligomers, but the AFM images do not appear to resolve the subunits. I think the authors should explain how they estimated the number of subunits.

3. In Fig. 2c, where the initiation time is discussed, how was $t=0$ defined? I think we can only quantify this about the initial time if the entire observation solution is sufficiently equilibrated at pH 4. Could the authors give comments on how this is ensured?

4. I am not sure how to look at Fig. 4. What is the difference in the color of each line? I also do not understand the relationship between the indications (e), (f), (g) and the lines. I want the authors to give easier and explicit explanations.

5. I also do not understand Fig. S4. Where can I see that it is a complementary electrostatic potential distribution?

6. In the main text Fig. S4 is mentioned after Fig. S1 before Fig. S2. I think it is better to match the order of the explanations and figures because it is confusing.

7. Experimentally, it seems that the breakage of the ring occurs in only one place in most cases, but what is the structural reason for this? I wonder what determines where within the oligomer the cleavage occurs?

General Comments to the reviewers:

To respond to the referee's comments, we have made the following major changes.

Figure 1: We rearranged the panels (reviewer #2), we labeled the distributions for clarity (reviewer #2), and added false color scale bars (reviewer #2).

Figure 2: We rearranged the arrowheads in the top row of the figure (reviewer #1 and #2), and added false color scale bars (reviewer #2). Amended caption, text and Methods for better clarity.

Figure 4: Clarifications regarding the line coloring (reviewer #1 and #2), the normalization of the lines (reviewer #1), and the labeling of the lines with regard to the structural assignments / salt bridge distances (reviewer #2) in caption and text.

Figure 5: Added false color scale bars (reviewer #2).

Figure 6: Added false color scale bars (reviewer #2).

Figure S1: Added false color scale bars (reviewer #2). Changed relative height axes to absolute heights over mica (reviewer #1).

Figure S2: Adjusted the contrast of the electrostatic potentials and highlighted the charge complementarities on the neutral pH interfaces with dashed lines (reviewer #2)

Figure S3: Added false color scale bars (reviewer #2).

Table S2: Added a new Supplementary Table S2 comprising all kinetic (and height) data derived from the HS-AFM experiments (reviewer #1).

In response to several comments by reviewer #1, we added several references and text parts discussing the cis- and trans- model of pore formation. We use the cis-model to interpret our data, mainly because we find it highly unlikely that pore-formation would occur into the bulk in our experiments, as would be expected based on the trans-model. We note however that, while our data is sufficiently resolved to assign the orientation of the pre-pore, we cannot exclude the trans-model of the pore state. However, the main achievements of this work are the detection of the clockwise hand-over-hand transition, the identification of a short-lived intermediate, and the determination of the kinetics of the pre-pore-I to pre-pore-II to pore transition – none of these aspects is challenged by the interpretational model.

RESPONSES TO THE REVIEWERS' COMMENTS

Reviewer #1 (Remarks to the Author):

Jiao et al present a time-resolved AFM study of Perforin-2/MPEG1, which represents one of three major mammalian immune effectors from the pore-forming MACPF/CDC superfamily. They conduct a biophysical analysis of the stepwise events during acid induced pore formation and in the process resolve key new intermediates. Remarkably, Jiao et al resolve a short-lived intermediate (half-life ~75 ms) that has not previously been identified.

Further, and distinct from prior studies, their work directly supports a clockwise propagation of the MACPF conformational change. The finding provides an important clue to the conserved shared mechanism of pore formation by MACPF/CDCs, with which seemingly disparate family members can now begin to be reconciled. For example, the Membrane Attack Complex is known to propagate in a similar clockwise fashion, however compared to Perforin and MPEG1 this appeared to be an exception (until now). Similarly, their data supports a model for arc pore formation and the recruitment of additional subunits, assuming the oligomerisation rate is sufficiently fast to add subunits to one end while the other is transitioning. (In fact these points may merit further comment in the discussion section).

Response: We thank the Referee for their overall positive general reception of our work. Commenting to the two points raised here that we will have the possibility to further elaborate.

With regard to the referee's comment: "...the Membrane Attack Complex is known to propagate in a similar clockwise fashion...". We assume the Referee refers to the following study: CryoEM reveals how the complement membrane attack complex ruptures lipid bilayers, Anaïs Menny, Marina Serna, Courtney M. Boyd, Scott Gardner, Agnel Praveen Joseph, B. Paul Morgan, Maya Topf, Nicholas J. Brooks & Doryen Bubeck, Nature Communications volume 9, Article number: 5316 (2018).

While these cryo-EM structures are, indeed, indicative of a transition process, it is per definition impossible to infer a process from still images. Only dynamic single molecule imaging can report about the nature of a process.

With regard to the referee's comment: "...the recruitment of additional subunits, assuming the oligomerisation rate is sufficiently fast to add subunits to one end while the other is transitioning." We think there might be a misunderstanding: There is no oligomerization addition process at work at the far end of the complexes. The complexes transition from preformed pre-pore assemblies. We do not consider a subunit recruitment process to be at work during the transition process.

The article is well written, comprehensive, and mostly the data support the conclusions (with some exceptions). I believe the article is suitable for publication in Nature Communications, with broad interest from the community. I remain sceptical about the interpretation of the data in the sole context of a cis-pore model, without entertaining alternative mechanisms, e.g. proposed by Pang et al.

Response: We thank the Referee for their comment about the writing and for supporting publication of our work. We agree that we must discuss alternatives to the cis-pore model as we have done in our former structural work.

In our revision, we have removed the clause "that likely implicates the flipping of the MACPF domain" from the description of the process in the introduction (lines 89 to 93). We have added an extensive paragraph (lines 161 to 170) in which we describe and discuss the cis- and trans- models and explain why we interpret our work in the framework of the cis-model.

We reiterate the differences between the cis- and trans- models in the results section (lines 277 to 278). We point out that the main objective of this work is the kinetics of the transition process, and not the structural interpretation. However, we also direct the referee to a paper which has now been placed on bioRxiv and has been submitted for journal peer review: <https://www.biorxiv.org/content/10.1101/2022.06.14.496043v2>. This paper provides further evidence for cis pore formation.

Major contributions.

- Direct evidence of a propagation of unfurling is a remarkable achievement and provides new insight into the conserved mechanism of MACPF/CDCs more broadly.
- The observation of a short lived prepore intermediate is very novel, suggesting a structural rearrangement must occur to drive the propagation of TMH unfurling. In the context of perforin, MAC and other family members this may be driven simply by oligomerisation.
- Comprehensive quantitative kinetics of the MPEG1/PRF2 pore formation process, allowing future computational modelling of the process.

Response: We thank the Referee for their positive reception of our work. It is indeed the 200 ms temporal resolution imaging mode performance and the 2 ms temporal resolution line-scanning performance that allowed us to characterize the clockwise transition process and the intermediate state, respectively. Thus, it is nice to see how technical advances enable new

discoveries.

Major concerns.

The suggested mechanism of MACPF flipping seems highly implausible and not directly supported by the evidence in this study (which mostly related to kinetics and transition features). The authors suggest ring breaking accommodates the rotation of the MACPF domain in order to re-arrange into a cis orientation, however this does not explain how the next MACPF domain can rotate given that this would still encounter steric hindrance, i.e. enormous main-chain clashes during rotation in the context of the oligomer.

Response: We agree with the Referee that the MACPF flipping mechanism (cis-model) is not directly supported by our HS-AFM data, and we have rewritten certain parts of the manuscript to clarify this (lines 89-93, 161-170, 277-278). On the other hand, the cis-model appears like the best fit for our dynamic HS-AFM data (increased height in the intermediate state and attack of the underlying membrane) and our previous cryo-EM results (see Figure 1 in the current manuscript). With regard to “**how the next MACPF domain can rotate**”, which is of course a very relevant question, we must refer to our molecular dynamics simulations (MDS) – this is why we engaged in this endeavor. As indicated by MDS, the CW exposed subunit is intrinsically unstable showing large fluctuations (Fig.3) and the inter-subunit salt-bridges with the next CW are destabilized (Fig.4). Thus, we imagine that we are looking at a chain-reaction of inter-subunit surface breakages, fluctuations, and conformational transitions of ‘almost isolated’ subunits in the CW direction (lines 287 to 290). It will be interesting if a flipped MACPF pre-pore structure could ever be resolved – maybe with the aid of a mutation blocking a later step of the transition. Finally, we point again to the paper now available on bioRxiv (<https://www.biorxiv.org/content/10.1101/2022.06.14.496043v2>) which proposes a possible twisting mechanism for ring breakage and for the steric enabling of MACPF domain rotation.

Furthermore, the suggestion implies that a stable MACPF/MACPF interactions must be broken, only to reform in an identical (yet rotated) state – which then somehow become unstable driving a conformational change. Why then would they not trigger in their original orientation? The energetics of this process seem highly unlikely, with the alternative explanation being far simpler, namely that the MACPF domain merely unfurls consistent with all other MACPF/CDC family members.

Response: Our explanation regarding the conformational intermediates must have been imprecise. The MACPF domain of the subunits do not reform – after rotation – in an identical (yet rotated) manner. Of course, the structure after MACPF domain flipping, i.e., pre-pore-II, is something entirely different from pre-pore-I. Also, we never claim (or should have made clearer that we do not think) that they reform a meaningful interface in pre-pore-II. First, we have to restate that the conformational change occurs at low pH, where titratable sidechains are protonated, and, thus, the interfaces between the subunits are weak (as for pre-pore-I). As we briefly explain above, our MDS results provide the following picture of the inter-subunit surfaces: 1) At acidic pH inter-subunit salt bridges are fragilized (see MDS Figure 4). 2) As a consequence, inter-subunit breakages occur (this is also supported by HS-AFM observation, e.g. Figure 2a,b). 3) Once there is an inter-subunit breakage, the subunit that is located CW is destabilized, i.e., exhibits strong structural fluctuations (see MDS Figure 3d). We do not know what the physical basis for this is, but it likely stems from the more fragile next inter-subunit salt bridges CW (Figure 4c,d). 4) As a consequence, the subunit in question stands quasi alone, and does the unexpected MACPF flip. 5) At this point, this subunit has obviously no meaningful interface with any of the two neighbors. However, now we can go back to step 3 for the next CW located subunit, and so on...

We also do not know if pre-pore-II makes any reasonable interactions with each other, but we do not think so. As briefly stated above, pre-pore-II is different from pre-pore-I, and there is

no reason to assume that a protein in which one domain out of several flips would be able to make meaningful interactions with its kind. Also, we do not think this is the case, because we know that pre-pore-II is only short-lived and there is no interest for the system to potentially get stuck in a pre-pore-II ring. Thus, pre-pore-II likely stands alone for a short time and then transforms into a pore. Obviously, pore subunits make meaningful and strong interactions again.

In brief, we do not think that after flipping the subunit interaction is reformed (and is certainly not identical). The conformational change to pore is only energetically favorable if a membrane can be attacked and the transmembrane beta strands find a membrane, which they only do after the rotation, according to the cis-model and in our experimental setup. Clearly the membrane-embedded state is a low-energy state. Likely pre-pore-II is a high-energy transition state, and that makes sense since it is only short-lived (lines 330 to 333).

We understand why the referee prefers a model for pore formation in which the MACPF TMH regions simply unfurl without domain rotation. However, the pore state structures (Ni et al 2020, <https://www.biorxiv.org/content/10.1101/2022.06.14.496043v2>) are rather supportive of a pore formation model in cis via MACPF domain rotation.

While the authors rely on the observation of a cis-like pore from cryoEM studies to support this model, this reviewer is not convinced. What if this reconstruction reflects an artefact, having been assembled in the presence of detergent and without a membrane bilayer to support and stabilise the P2 domain? Indeed, this study itself supports the notation that the P2 domain becomes highly mobile upon acidification and thus could simply be disordered in the imaged material. The proposed mechanism raises more questions than it solves. I believe these issues could be rectified by interpreting the AFM and computational data in light of other possible mechanisms as well.

Response: The cryo-EM structure is an average over hundreds of thousands of molecules and it seems highly unlikely (unheard of) that all these particles that average to a consensus with some angstrom resolution would all be artefacts. Also, in the AFM experiments (see former paper) we have only one membrane, and the proteins transit into a pore state with dimensions and heights compatible with the cryo-EM structure. It appears highly unlikely that the protein would unfurl its transmembrane beta strands into the bulk and/or create the same kind of artefact in two structural methods, one using membrane and physiological environment. As detailed above, we discuss in revision the cis- and trans- models in better detail. We also point out that this manuscript is entirely a HS-AFM work, and only uses the cryo-EM structures for interpretation, but we do not want our manuscript to be judged based on disagreements between cryo-EM works.

Other concerns.

- Fig 1, p4: I found the height measurements of the pore to be confusing and inconsistent with other reports. Ni et al measure the full pore height (including transmembrane region) to be only 14 nm based on cryo-EM (PDB 6SB5), where the transmembrane region accounts for ~4 nm (consistent with bilayer width) leaving ~10 nm above the membrane. Together these are consistent with the measurements on mica (Fig 1j), but not the measurements on membranes which gives a taller assembly at 12 nm above membrane (Fig 1l) or 16 nm when including the membrane (Fig 1f), i.e., a difference of between 1.5 to 2 nm. I suppose a difference of 2 nm may be within experimental error? Couldn't these data also be consistent with a trans-pore as suggested by Pang et al? These would sit roughly 8 nm above membrane as a prepore and roughly 14 nm as a pore, give or take depending on the behaviour of the transmembrane region without a target bilayer to stabilise it. I note that Ni et al measure the prepore and pore height in their previous report at 4 and 8 nm respectively (fig 5) – I assume this discrepancy is due to an incorrect reference point i.e. miscalibration. In this study, how were the heights calibrated?

Response: Let us recall the measurements and discuss briefly where things are not in

complete agreement.

In Ni et al. 2020:

cryo-EM pre-pore height: 8.3 nm

cryo-EM pore height: 14 nm (loops may be missing); interpreted: 11 nm + 3 nm in membrane

In the current work:

on mica: HS-AFM pre-pore height: 8.2 nm

on mica: HS-AFM pore height: 14.0 nm

on membrane: HS-AFM pre-pore height: 8.1 nm

on membrane: HS-AFM pore height: 12.1 nm (above membrane)

on membrane: HS-AFM pore height: ~16 nm (next to membrane defects, Fig. S1b)

Δ -height ~4 nm from pre-pore to pore: 4 nm to 8 nm (in Ni et al. 2020, Fig. 5)

In summary, all measurements precisely agree that the pre-pore is ~8.2 nm.

When measuring the entire pore thickness, cryo-EM might miss some loops above the membrane. HS-AFM on the other hand might measure an additional 1 nm below the membrane, between the membrane and the mica. Thus, when we measure ~16 nm from top of the molecules to the mica, this likely comprises a 1-nm buffer layer between the sample support and the membrane. Thus, the pore is likely 12 nm (above the membrane) + 3 nm (in membrane). Overall, the data fit rather well. We integrated all molecular heights in the new Table S2, in which we also gather the kinetic data as per the referee's request.

The Delta-height from pre-pore to pore is ~4 nm in good agreement with 8.1 nm to 12.1 nm (above membrane). We regret though the left-hand scale in Figure 5 of Ni et al. We should have used a relative Delta-height scale bar instead. Indeed, while the Delta-height measured there is correct, the HS-AFM tip could not reach all the way to the membrane level in between the densely packed molecules – our bad in 2020.

In our revision, we have added to all our images false color scale bars and adjusted the heights in the graphs to absolute values above mica, instead of relative scales to avoid future confusions as have obviously emerged from Figure 5 of Ni et al. 2020.

We again acknowledge the referee's preference for pore formation in trans. We do not say this cannot occur, but we do say that all pore structures determined so far, whether in isolation or on a membrane, indicate instead pore formation in cis, and the AFM experiment is much more compatible with this interpretation too.

- Line 215, p6, Figure 2 & MovieS2: Prepore opening/breakage is not obvious to me (perhaps untrained eye). This could be made clearer, perhaps a rotational average of prepores in the split washer (open) state? It is also not clear to me how these opening/breakages were detected and quantified. Were they counted manually? This could introduce bias.

Response: In our revision, we have labeled openings with oriented arrowheads in the first row of Figure 2a. We have amended the caption of Figure 2 accordingly. The pre-pore ring breakages were counted visually, as defined in Methods, Determination of kinetic parameters, Pre-pore-I inter-subunit breakages: (~Lines 458-464).

We estimate the rate to ~1.8 s⁻¹, but do not draw major conclusions from this number. We note that breakages occur rather frequently at acidic pH (and not at neutral pH), but closure of breakages is much faster (>>5 s⁻¹), and that is why most often only one transition per ring occurs.

- The suggested mechanism implies arcs would be more prone to transitioning, since opening is key to the transition of prepore to pore. Given that PRF2 assemblies on mica were typically arcs, do you see a faster transition time for these assemblies relative to complete prepores? Similarly, I would predict pre-assembled prepores placed on mica would have a reduced rate of transition, since mica would restrict the opening dynamics.

Response: We did not measure the transition rate of PFN2 pre-pores to pores on mica. As mentioned by the Referee, the mica surface could likely interfere with the transition process due to electrostatic interactions. Anyway, we agree with the reasoning that pre-pore arcs

should perhaps be more prone to transitioning than full rings, though the opening rate of 1.8 s^{-1} of full rings is also high. On membranes at the concentrations used in AFM measurements most of the pre-pore assemblies are full rings, and we cannot comment on the action of the few pre-pore arcs. When pores are formed by adding perforin-2 to vesicles the structures formed are mostly arcs (<https://www.biorxiv.org/content/10.1101/2022.06.14.496043v2>).

- Line 225, p8: The MD analysis suggests the CTT and P2 domains are most affected by lowering the pH (this domain is also highly mobile in cryoEM, observed by both Ni et al and Pang et al), however here the authors suggest it is the MACPF domain which undergoes a 180° flip. I don't think this is supported by the simulations. The MACPF domain has approx. constant RMSD (Fig S2c,d). Have the authors tried more sophisticated analysis, such as PCA to see what domain movements capture the largest variation components of the data? Perhaps the time scales are insufficient to see early signs of flipping, but do these show salt bridge breaking?

Response: The molecular dynamics simulations (MDS) were not conducted to get any information about the MACPF flipping, likely to span timescales not amenable to computer

simulations, nor would one expect that MDS could ever predict or reproduce a large conformational change without steering – also calling into question the meaning of a single realization. The MDS were only performed (1) to assess why the CW located subunit is more prone to undergo a conformational change, and (2) to assess how inter-subunit salt bridges weaken upon acidification, leading to structural fluctuations within the pre-pore ring conducive to any sort of conformational changes.

We scrutinized the entire manuscript and carefully removed every possible notion that MDS would be in favor or disfavor of the cis- model of membrane attack. Let us elaborate again why we favor the cis-model with the MACPF domain flipping. The assignment of the topography of the pre-pore structure with respect to the membrane is unambiguous: As the Referee can see, the pre-pores on the membrane in (e) have an innermost diameter of the top protruding ring of $\sim 11 \text{ nm}$. This assigns the orientation of the pre-pore with respect to the membrane, as illustrated in (b), where the CTT face the membrane (as expected) and the TMHs (red) face upwards and the MACPF (blue) is below the TMHs. After the transition (i), clearly the TMHs are in the membrane (f, bottom). Thus, the MACPF domain must have flipped somehow to bring the TMHs towards the membrane, as certain other parts of MACPF (see the two helices) remain essentially in their pre-pore fold but flipped. Since the penetration of the TMHs into the membrane pushes the entire structure upwards, it is not very surprising that the CTT is not anymore resolved in the pore conformation - it likely hangs there in the bulk underneath P2, wobbling around. In contrast, in the pre-pore state, it is stabilized by the membrane interaction. If anything, then the MDS suggest that the increased flexibility of CTT at low pH favors its detachment from the membrane as is likely required for the conformational change that is to follow.

We understand that the trans- model as suggested by Pang et al. has appeal because it does not implicate the flipping of the MACPF domain. From the perspective of our HS-AFM experiments only, we can hardly imagine that the protein during the pore transition would

unfurl its TMHs into a pore into the bulk, as we do not have a second trans- positioned membrane. Therefore, we interpret our data in light of the cis- model.

- Line 272, p8: Generally, the MD analysis provides a nice framework to understand the pH sensitivity of PRF2. However, I feel this conclusion would be better supported by mutagenesis and activity assays e.g. red blood cell lysis or dye leakage assay. The authors should consider targeting key salt bridges with alanine substitutions to determine whether these prepores become activated at more mild pH, e.g. D448A, E99A and K332A. Specifically, if these salt bridges are indeed critical, their loss should render the protein more active in less acidic conditions or at least increase the opening rate.

Response: These are, indeed, excellent suggestions, and we agree that the MDS would make such predictions. Obviously, it will not be that straightforward to predict what consequences such interface mutations will have though...

- Line 281, p9, fig 4a-d: What does each line represent in the probability distribution curves? Are these individual salt bridges? What is the unit of the vertical axis, a fraction of 0 to 1 or percentage? Many values exceed a “probability of 1”.

Response: Each line represents an individual salt bridge. It is a normalized probability, where each salt bridge’s length over the simulation time is pooled giving a fixed integrated occurrence for each salt bridge: $\int dx h(x) = 1$ (see orange and indigo dashed lines that are the integrals of the red and the cyan lines / salt bridges). We have amended the figure caption accordingly.

Minor issues.

- Line 82, p2: “Previous studies suggest PFN2 can form pores in a second” – the phrasing is confusing, would suggest “Previous studies suggest PFN2 can form pores on time scales of one second”. This prevents confusion with second membranes, as suggested in Pang et al.

Response: We thank the Referee for the recommendation for improvement. We have rephrased the sentence to now read “Previous studies suggest PFN2 can form pores on time scales of one second”.

- Line 81, p2: “dynamics remain largely unknown” - The authors should mention the studies of Parsons et al (2019, Nat Commun) and McGuinness et al (2022, bioRxiv) which both address similar topics of MACPF/CDC dynamics – especially since these reports are consistent with and further augment the findings herein (with respect to arcs and clockwise propagation of insertion).

Response: We disagree with the Referee. The quote “dynamics remain largely unknown” appears like the right thing to say. We certainly should cite the Parsons et al. publication, but prefer to wait for the McGuinness et al. work to be peer-reviewed and published before citing it (we also do not cite the above-mentioned non-peer reviewed structure work from some of the authors here). With respect to the Parsons et al. publication, we point out the Methods employed: “Imaging was generally performed in off-resonance tapping/fast force-feedback imaging (Bruker’s PeakForce Tapping) mode where force-distance curves were recorded at either 8 or 32 kHz, with amplitudes of 10–20 nm. With these frequencies, images could be collected at 5–100 s/frame.” For example, the oligomerization image in Figure 4 in Parsons et al. was taken at 6.5 s/frame (see its figure caption) – and does not show CW oligomerization. In contrast, our pre-pore to pore transition imaging was performed at 200 ms/frame, thus 32.5 times faster. Understandably, we cannot compare these results and the meaning of them

remains unclear to us. Also, we disagree that this paper shows CW oligomerization, nor that the paper ever claims to show that. The paper mentions the sequence of protein units that are assembled in the MAC complex, and then, based on this sequence and referring to cryo-EM papers, the word “clockwise” is mentioned just once (in the introduction) – assuming that the biochemical sequence and the structural elements indicate chronology (the AFM data resolves c5b by height, but the data does not resolve any of the other units nor the directionality). With regard to the referenced cryo-EM data, e.g., Menny et al. Nature Comm 2018, we agree that the data is suggestive of a pathway, but it does not show propagation. Static data can be interpreted in dynamic ways, but it does not represent primary data that constrain dynamics or chronological order of events.

We have rather strong feelings about this, as it is our ambition and a complicated effort to further develop a dynamic structural technique to answer such questions. We find it sometimes difficult when claims about dynamics and chronologies are made from static images.

- Line 103, p3: For completeness, please in parentheses state the alternate name for the P2 domain i.e. MABP.

Response: We thank the Referee for this suggestion and have amended the sentence by adding “(also termed MABP)” (Line 104, p3).

- Line 121, p3: Nor was it resolved in the membrane bound prepore (Pang et al). As such fig1b is inaccurate as the CTT is shown.

Response: The CTT domain was resolved in the cryo-EM pre-pore data, see Figure 1f,j in Ni et al 2020). What we say in line 121 is that the CTT domain was not resolved in the pore state. In the pre-pore formed on a membrane the CTT remains partly resolved, for mPFN2 up to residue 628 (<https://www.biorxiv.org/content/10.1101/2022.06.14.496043v2>). This supports the depiction made here but see below for further discussion.

- Line 151, p4: Figure 1b – Ni et al did not determine a membrane bound PRF2/MPEG1 structure; PDB 6SB3 is a soluble prepore, whose orientation is inferred. The PDB 6U2W from Pang et al should at least be included.

Response: We amended the text to refer to Pang et al. (Line 105, p3). We agree that a strong assignment of sidedness can only be done from a structure on a membrane. However, the orientation of the pre-pore structure with regard to the membrane (also in Ni et al. 2020) appears unambiguous based on the HS-AFM topography (the ~11 nm top-ring diameter of the bulk-exposed pre-pore surface, compare figure 3a,b,c and figure 1b in Ni et al 2020).

- Line 67, p2: First instance of acronym “PRF2” in the main text, please also include the other common name, “MPEG1”. Fine for the rest to be only PRF2.

Response: We use PFN2 (not PRF2) and have added “MPEG1” when first defining the abbreviation of Perforin-2 in the first sentence of abstract.

- I found interchanging between terms (i.e., lifetime, half-life, transition/initiation time, time constant, etc) somewhat confusing. I realise these are standard terms, perhaps to improve interpretability it would benefit the reader to have a summary table which includes all these quantities (including uncertainties) in one place.

Response: We went through the manuscript and tried to use identical terms whenever possible. We highlighted all terms in the marked version of the revised manuscript. We also summarize all kinetic data in a new supplementary Table 2. Many parameters are also featured in the summary figure (Figure 7).

New Table S2:

Pre-pore-I height	8.2 ± 0.4 nm (on mica) 8.1 ± 0.2 nm (above membrane)
Pre-pore-I inter-subunit opening rate (at neutral pH)	Not observed
Pre-pore-I lifetime (at neutral pH)	>> observation time
Pre-pore-I inter-subunit opening rate (at low pH)	1.8 ± 1.2 s ⁻¹
Pre-pore-I inter-subunit closing rate (at low pH)	> 5 s ⁻¹
Pre-pore-I lifetime (at low pH)	5 s
Pre-pore-II height	~9.7 nm (above membrane)
Pre-pore-I to pre-pore II Δheight	1.6 ± 0.4 nm
Pre-pore-I to pre-pore II conformational change height increase speed	47.3 ± 5 nm/s
Pre-pore-I to pre-pore II conformational change time	~34 ms
Pre-pore-I to pre-pore-II transition rate	~13 s ⁻¹
Pre-pore-II lifetime	~75 ms
Pore height	14.0 ± 0.5 nm (on mica) 12.1 ± 0.6 nm (above membrane) ~16.0 nm (above mica in membrane)
Pre-pore-II to pore Δheight	2.2 ± 0.2 nm
Pre-pore II to pore conformational change height increase speed	48 ± 7 nm/s
Pre-pore-II to pore conformational change time	~46 ms
Pre-pore-II to pore transition rate	~15 s ⁻¹
Pre-pore-I to pre-pore-II to pore total transition time	1.3 ± 0.6 s
Pore lifetime	>> observation time

- Please include height scale colour bars in all figures e.g. figs 1d, e, h, i; 2a; 5a; 6a.

Response: We have added z-scale false color bars in all figures (Fig. 1d, e, h, i; Fig. 2a; Fig. 5a; Fig. 6a; Fig. S1a, b; Fig. S3a, b).

- Many quantities (particularly in discussion and methods) are given without a space between the number value and unit, e.g. line 392, p12: “1.8s-1”.

Response: We have added spaces between number values and units throughout the manuscript – all highlighted in the marked-up version of the revised manuscript.

- Line 193, p5: “quite conserved” this term seems more appropriate for evolutionary processes, perhaps “quite consistent” would be clearer.

Response: We have amended to “quite consistent”.

- Temperature is not stated in the methods or main text, these kinetics will be temperature dependent and as such it should be stated alongside the kinetic rate parameters determined.

Response: We define “at room temperature” both in the methods (twice) and main text.

- Perhaps this is beyond the scope of the study, but the authors should consider performing similar analyses using the disulphide trapped proteins reported in Ni et al. For example, is the prepore II intermediate still observed with PRF2K251C/G286C? Something to think about.

Response: We agree that the disulphide trapped proteins with PFN2 K251C/G286C would be an interesting target to investigate regarding the pre-pore-II intermediate state. We probably will perform such experiments in the future, after further experiments addressing the cis- vs trans- model.

- Line 246-248, p7: “This result is noteworthy, because the CTT makes membrane contact in the prepore state ...” – this is inaccurate, the CTT is not resolved in the membrane bound prepore reported by Pang et al owing to conformational changes of the P2 domain β -hairpin upon membrane binding.

Response: As we mentioned in the above question, we referred to the CTT in the cryo-EM data shown in Ni et al 2020. To discuss this point in greater detail, we have made a composite of the structure figures – Figure 1j in Ni et al 2020 (left) and figure 2d in Pang et al. 2019 (right). According to Ni et al. the membrane-facing side is downwards in this image. According to Pang et al., the membrane is indicated and on the same side (though we had to rotate the panel). The two structures are essentially identical. We guess, the discussion emerges from the fact that the short helix resolved in Ni et al. is termed CTT, while Pang et al. actually also depicts this short helix. The controversy arises from Ni et al. preferring the cis- model, based on the structure of a PFN2 pore, where the MACPF domain flips and attacks the membrane below, while Pang et al. prefer a trans- model where the TMHs unfurl upwards attacking another membrane. As discussed, we found it unlikely that the complex unfurls a transmembrane beta-barrel towards the bulk in our experiments, but cannot exclude it. Therefore, we discuss the trans- model in our revision, but point out that the interpretation would not change the key findings of our paper - the clockwise pre-pore to pore transition and the kinetics of all states. We think that this illustration resolves several of the above discussed points. It is also worth noting that the precise configuration of the P2 domain hairpin tip and the CTT seems to depend on the order of assembly of pre-pores with the membrane. See <https://www.biorxiv.org/content/10.1101/2022.06.14.496043v2> where pre-pores assembled on a target membrane do resolve the CTT helix (depicted in the composite figure accompanying this section of our response, like in Pang et al. 2019) interacting with the surface of the lipid bilayer. But clearly, further experiments are needed to work out the discussion regarding the cis- and trans- models.

- Line 582, p17: “perforin” should be “perforin-2”

Response: We have amended the “perforin” with “PFN2”.

- Line 283, p9: “and acidic (b,d) pH” extra parentheses, should be “and acidic (b,d) pH”.

Response: We have amended the sentence to now read “and acidic (b,d) pH”.

Congratulations on a beautiful study,
Charles Bayly-Jones

We thank Dr. Charles Bayly-Jones for his thorough and detailed review. We understand that the Referee is not convinced by the cis- model of membrane attack, and we hope our line of argument assuaged his concerns. We further hope that this exchange will help the involved groups to parse out the topic and provide better experiments in the near future. Anyway, the major conclusions of this work, the CW transition, the intermediate state, and the entire set of kinetic parameters are not dependent on the structural interpretation, but stand solid on their own. But we find that we cannot take another preferred line of interpretation than the one we have in the cis- model, given the pore structures we have observed.

Reviewer #2 (Remarks to the Author):

Jiao et al. have successfully captured the pH-dependent structural transition from the pre-pore to the pore state and its intermediates in the membrane pore-forming protein Perforin2 using the latest high-speed AFM techniques. Based on the HS-AFM imaging, they revealed counterclockwise conformational transition along the ring-shaped oligomer and also quantified the kinetics of the structural intermediates. Further the MD simulations have also provided a molecular structural understanding of the clockwise unidirectional conformational transition, which is quite convincing as a mechanistic explanation of the experimental facts. These results effectively demonstrate the potential of HS-AFM technique and contribute greatly to our understanding of the mechanism of pore formation in membrane pore toxins. Thus I consider that this work is worthy of publication in Nature Communications. On the other hand, I felt that the manuscript was not well organized and the explanations of some figures were quite difficult to understand, which made me stressful. I will make some comments below, which I think would make the manuscript better if revised.

Response: We thank the Referee for their positive general reception of our work. We regret that the Referee found some parts of our manuscript disorganized, and we have amended it according to their recommendations.

1. First of all, "pre-pore" and "pore" in Fig. 1 are very confusing. I think the authors refer the structures of PFN2 observed at the different pH conditions of the solution as "pre-pore" and "pore", which made me really confusing at first. Also, because of the layout of the figures, I mistakenly thought that Figs. j and k were the results of the analysis of Figs. d and e. It would be better if the definitions of "pre-pore" and "pore" is clarified at the beginning of the manuscript, and the layout of the figures were also reconsidered.

Response: Pre-pore always refers to the structure and state when the protein sits on the membrane, while pore always refers to the structure and state when the protein has inserted into the membrane and makes a transmembrane pore. Indeed, it is confusing that panels j,k and l,m do not directly relate to panels d,e and h,i. Indeed, these histogram panels contain mixed information from various images. In our revision we have changed Figure 1 and moved the histogram panels below the correctly related panels pre-pore (b,c,d,e) and pore (f,g,h,i), and labeled the distributions in (j, k, l, m) with the HS-AFM data panel from which the distributions are derived. We believe Figure 1 is much clearer in the amended version.

1. I understand that the difference in brightness in the AFM image in Fig.d-i is due to the difference in height, but it would be easier to understand if a color bar could be added next to each image.

Response: We agree and added z-scale false color bars to all panels (Figure 1d,e,h,i; Figure 2a; Figure 5a; Figure 6a; Figure S1a,b; Figure S3a,b). In addition, we adapted relative height scales to correspond to absolute heights over mica, as also Referee #1 pointed out in one comment that relative height scales might lead (or have led in a former paper) to confusion.

2. Several analyses account for the number of subunits of oligomers, but the AFM images do not appear to resolve the subunits. I think the authors should explain how they estimated the number of subunits.

Response: The Referee is right. The number of subunits were calculated by fitting the radial length of the pre-pore and pores structures, knowing that the full ring corresponds to 16 subunits. Amended in Materials and Methods (Lines: 476-478).

3. In Fig. 2c, where the initiation time is discussed, how was $t=0$ defined? I think we can only quantify this about the initial time if the entire observation solution is sufficiently equilibrated at pH 4. Could the authors give comments on how this is ensured?

Response: The Referee is right. It is essentially impossible to know when all the solution is equilibrated at pH 4. Therefore, $t=0$ was defined as the moment, when the first pre-pore ring initiated a pre-pore to pore transition. Given that the time constant of the initiation is slow, ~ 5 s and the frame rate is 0.2 s, we are sure that we capture enough data of the exponential decay of the ensemble pre-pores to assess a meaningful time constant. We have amended figure caption 2a, accordingly.

4. I am not sure how to look at Fig. 4. What is the difference in the color of each line? I also do not understand the relationship between the indications(e), (f), (g) and the lines. I want the authors to give easier and explicit explanations.

Response: We are sorry that we did not specify the color of the lines. They simply illustrate individual salt bridges, and making them in different colors rendered the visibility better than having all lines in the same color. Labels e, f and g represent salt bridges of different lengths, those that are direct (e) are at short distances, namely ~ 3.5 Å, those that are indirect and water molecule mediated are at ~ 6 Å (f) and longer, ~ 8 Å (g), where the labels in the top panels correspond to examples in panels e,f,g. We have amended figure caption 2 accordingly.

5. I also do not understand Fig. S4. Where can I see that it is a complementary electrostatic potential distribution?

Response: The Referee is right. We have amended Figure S4 (now reorganized as Figure S2) so that the electrostatic potential colors (red, blue) are stronger (greater saturation and enhanced gradient), and oriented the subunits for better visibility. We have also added dashed outlines to highlight the charge complementarities on these surfaces.

6. In the main text Fig. S4 is mentioned after Fig. S1 before Fig. S2. I think it is better to match the order of the explanations and figures because it is confusing.

Response: We thank the Referee for pointing this out. In our revision, we have reorganized the order of Supplementary Figures, Figure S4 is now Figure S2.

7. Experimentally, it seems that the breakage of the ring occurs in only one place in most cases, but what is the structural reason for this? I wonder what determines where within the oligomer the cleavage occurs?

Response: Regarding the location of the breakages, we have used only one arrowhead to label all ring breakages, but these arrowheads did not really point to the real-space locations of the breakages. The breakage sites are due to fluctuations between subunits in the acidic environment and are randomly distributed. In response, we use differently oriented (and more) arrowheads to more precisely point out the breakages. Regarding the occurrence of the breakages, the opening rate of pre-pore ring breakage is estimated to ~ 1.8 times per second, but the closing rate is much higher, $\gg 5$ s⁻¹. Accordingly, the probability of breakage in each ring is most likely to occur only in one place inducing pre-pore to pore transition. In response, we have amended the arrowheads of Figure 2, thereby clarifying the location and occurrence of the breakages.

Reviewers' Comments:

Reviewer #1:

Remarks to the Author:

Overall, the article is improved, and the authors have responded to all my comments and suggestions. There remain some minor points of confusion and comments that I feel have not been addressed.

a. "...the Membrane Attack Complex is known to propagate in a similar clockwise fashion...". Please see Parsons et al. (e.g. figure 4) – the original comment remains relevant. The similarities in unidirectional assembly/pore formation between MPEG1 and MAC merits some discussion, even if there remains further single-molecule imaging to address the outstanding questions of MAC pore formation.

I appreciate the authors feel strongly about claims of dynamics from static imaging – however MAC chronology has been studied by various groups with consistent findings. The sequential and unidirectional assembly of MAC is therefore quite well established. It is supported by QCM-D (Parsons et al. [2019], Yorulmaz et al [Biomacromolecules, 2015]), cryoEM (Serna, Spicer, Menny, etc), AFM (Hoogenboom), and red cell / liposome leakage assays (Spicer et al. [2018]).

b. "...the recruitment of additional subunits, assuming the oligomerisation rate is sufficiently fast to add subunits to one end while the other is transitioning." I understand (and agree with the author) that the oligomerisation process is not occurring in these in vitro AFM experiments. However, in a physiological context MPEG1 monomers may be delivered directly to a low pH environment. Indeed Ni et al. (2020) show that low pH induces oligomerisation, suggesting that oligomerisation occurs in the phagolysosome/late endosome. Thus, both processes (oligomerisation and transition into the pore state) may commence or occur together in time. Since oligomerisation is much slower (2.4 s^{-1} [Ni et al]) than the transition process ($\sim 13 \text{ s}^{-1}$) these data suggest that prepore oligomers may not have time to fully assemble, thus preferencing the formation of arcs.

c. The MDS indeed shows large fluctuations, however these are predominantly in the CTT and P2 domains. The MACPF domain appears equally stable in both the CW and CCW subunits at neutral and low pH. See blue curve in Fig S4.

d. There remains confusion about the structural data of the CTT region (referred to as the L-domain in Pang et al). This region is not resolved in the 3.9 \AA membrane-bound prepore structure (PDB 6U2W; Mpeg1 bound to liposomes). The figure (2d) has been misunderstood, it depicts a superposition of the soluble (green) and membrane bound (grey) structures. I notice in the mentioned preprint that the L-domain/CTT can be resolved in murine MPEG1.

Reviewer #2:

Remarks to the Author:

The authors gave clear answer to my questions and improved the manuscript very much. I am now satisfied with all revision.

Reviewer #3:

Remarks to the Author:

The paper describes a potential mechanism for the pre-pore to pore transition of Perforin-2. AFM experiments and MD simulations hint at a clockwise hand-over-hand process. The paper is generally well-written and clear and the proposed mechanism is reasonable.

The experiments and general biology of the paper have been discussed in other reviews, so I will focus on the MD simulations, where several points remain unclear:

1. The methods section does not mention modeling the lipid membrane in their simulations. Thus, I assume the trimers were simulated in water only which is highly unusual for a membrane-active protein. How is this omission of a critical component of the overall system justified? Valid conclusions might be drawn even from simulations without membrane but it should be explicitly discussed why the authors think this is the case here.

2. The results figures indicate that for each condition, only one simulation run was performed. Is this indeed the case? It is not guaranteed that a single trajectory is representative of the general behavior. A single transition might happen by chance in one condition but not in the other one, even if the differences between them are not significant.

3. The CTT domain of CCW subunit-a at pH 4 appears to adopt a distinct (possibly metastable) state (Figure S4) but the paper does not show what this state looks like. It would help to show representative simulation "snapshots" to compare its behavior at pH4 and pH7.

4. The fluctuations of CW subunit-c (figure 3) do not appear to vary significantly between pH7 and pH4. It looks like P2 and CTT appear to deviate a bit more from the starting structure (Figure S4). As mentioned in issue 2, in a highly dynamic system, this could be a mere coincidence if observed only in one simulation run (that is relatively short compared to the timescales of the physical process). As in the above point, structural snapshots would be helpful in assessing the conclusions made in the paper.

Minor issues:

- Figure S2: In the caption, the second occurrence of CCW in "CCW (a) and CCW (b)" should be CW, I presume.

RESPONSES TO THE REVIEWERS' COMMENTS

Reviewer #1 (Remarks to the Author):

Overall, the article is improved, and the authors have responded to all my comments and suggestions. There remain some minor points of confusion and comments that I feel have not been addressed.

Response: We thank the Referee for their overall positive general reception of our work.

a. "...the Membrane Attack Complex is known to propagate in a similar clockwise fashion...". Please see Parsons et al. (e.g. figure 4) – the original comment remains relevant. The similarities in unidirectional assembly/pore formation between MPEG1 and MAC merits some discussion, even if there remains further single-molecule imaging to address the outstanding questions of MAC pore formation.

I appreciate the authors feel strongly about claims of dynamics from static imaging – however MAC chronology has been studied by various groups with consistent findings. The sequential and unidirectional assembly of MAC is therefore quite well established. It is supported by QCM-D (Parsons et al. [2019], Yorulmaz et al [Biomacromolecules, 2015]), cryoEM (Serna, Spicer, Menny, etc), AFM (Hoogenboom), and red cell / liposome leakage assays (Spicer et al. [2018]).

Response: We agree that all structural snapshots are indicative that the C9 protein binds to the C5b-8 to propagate clockwise oligomerization. However, it is important to discern assembly/oligomerization and conformational transition. Clearly, the assembly dynamics of MAC imaged by AFM (over 104s, at 6.5s/frame) in figure 4a in Parsons et al. do not resolve clockwise propagation dynamics, nor do they claim that they can. However, to respond to the reviewer's request, we added a sentence stating that static structures evidence a clockwise assembly of MAC (line 372-374).

b. "...the recruitment of additional subunits, assuming the oligomerisation rate is sufficiently fast to add subunits to one end while the other is transitioning." I understand (and agree with the author) that the oligomerisation process is not occurring in these in vitro AFM experiments. However, in a physiological context MPEG1 monomers may be delivered directly to a low pH environment. Indeed Ni et al. (2020) show that low pH induces oligomerisation, suggesting that oligomerisation occurs in the phagolysosome/late endosome. Thus, both processes (oligomerisation and transition into the pore state) may commence or occur together in time. Since oligomerisation is much slower (2.4 s^{-1} [Ni et al]) than the transition process ($\sim 13 \text{ s}^{-1}$) these data suggest that prepore oligomers may not have time to fully assemble, thus preferencing the formation of arcs.

Response: This is an interesting discussion. It is true that we cannot exclude the further addition of subunits to one end while the other end is transitioning. However, we do not think this is happening. Please note that the oligomerization rate that we described earlier in Ni et al. concerned only the addition of subunits to arcs in a densely packed environment and unlikely reflects the 2D diffusive lateral assembly, likely rather subunit exchange or recruitment of some rare still soluble protomers. Also note that the transition process of 13 s^{-1} is the transition rate within a preexisting assembly. Likely a more important number to consider is the transition initiation rate of 0.2 s^{-1} (line 369-371); pre-pores are rather stable, even at low pH. Therefore, we consider that the oligomerization and transition processes are

separated from each other, as in other PFPs.

c. The MDS indeed shows large fluctuations, however these are predominantly in the CTT and P2 domains. The MACPF domain appears equally stable in both the CW and CCW subunits at neutral and low pH. See blue curve in Fig S4.

Response: The CTT and P2 domains are binding the complex to the membrane in the prepore state. The fact that upon lowering the pH, the largest fluctuations were observed in these CTT and P2 domains is rather intuitive to activate the complex to transit into pores. The finding that the MACPF domain appears equally stable (blue curves in Fig S4) in the CW and CCW settings at both pHs seems rather unrelated to us. Indeed, the conformational transition must start somewhere, in this case in the parts that seem important to mediate the membrane bound prepore state, and then can propagate further. In brief, the MDS only evidence increased RMSD at pH4 as compared to pH7, and increased RMSD in the CW setting as compared to the CCW setting, but they are agnostic to what is going to happen next (line 243-246).

d. There remains confusion about the structural data of the CTT region (referred to as the L-domain in Pang et al). This region is not resolved in the 3.9 Å membrane-bound prepore structure (PDB 6U2W; Mpeg1 bound to liposomes). The figure (2d) has been misunderstood, it depicts a superposition of the soluble (green) and membrane bound (grey) structures. I notice in the mentioned preprint that the L-domain/CTT can be resolved in murine MPEG1.

Response: We apologize for the confusion. The CTT domain is partially resolved in the membrane-bound PFN2 prepore structure in the preprint (residues Ser600 – Lys628, compared to the solution prepore structure, PDB: 6SB3). As the reviewer can see in the preprint (see image below), orientation of the domains with respect to the membrane and the comparison of all structures is compatible with our interpretations and model (if anything, it reinforces the notion that P2 domain fluctuations may be important in the activation process of the transition – regarding comment (c) above).

From: Yu et al. 2022, bioRxiv preprint doi: <https://doi.org/10.1101/2022.06.14.496043>

E. Comparison of pre-pore PFN2 structures from the human homolog (pink, PDB: 6U2W), mPFN2 in isolation (gray, PDB: 6SB3) and mPFN2 formed on a membrane (green). Red arrows indicate the major differences among the structures.

Reviewer #2 (Remarks to the Author):

The authors gave clear answer to my questions and improved the manuscript very much. I am now satisfied with all revision.

Response: We thank the Referee for the reception of our revision.

Reviewer #3 (Remarks to the Author):

The paper describes a potential mechanism for the pre-pore to pore transition of Perforin-2. AFM experiments and MD simulations hint at a clockwise hand-over-hand process. The paper is generally well-written and clear and the proposed mechanism is reasonable. The experiments and general biology of the paper have been discussed in other reviews, so I will focus on the MD simulations, where several points remain unclear:

Response: We thank the Referee for overall positive reception of our work. We are grateful to the Referee for the constructive comments to help improve the quality of our manuscript and clarify important points therein.

1. The methods section does not mention modeling the lipid membrane in their simulations. Thus, I assume the trimers were simulated in water only which is highly unusual for a membrane-active protein. How is this omission of a critical component of the overall system justified? Valid conclusions might be drawn even from simulations without membrane but it should be explicitly discussed why the authors think this is the case here.

Response: The Reviewer is correct in their assumption that the trimers were simulated in water, as opposed to on top of a lipid bilayer. Our choice is justified by the several facts. First, the prepore structure is a solution structure. At the time of our work, we only had a prepore structure solved in solution at hand, only recently, we have been able to solve a prepore structure on membrane (, which is very similar. see: Yu et al. 2022, bioRxiv preprint doi: <https://doi.org/10.1101/2022.06.14.496043>). Second, the conformation of the perforin monomers that we examined in our molecular dynamics simulations is that prefacing the very large conformational transition, leading ultimately to the perforation of the membrane. In this prepore state, the self-assembled PFN2 structure sits at the surface of the bilayer, its monomers exposed to the aqueous environment. Third, the role of our molecular dynamics simulations is to rationalize why the conformational transition of PFN2 from pre-pore to pore is more likely to occur in a clockwise direction — not to investigate the actual conformational transition, which is beyond the capacity of enhanced-sampling strategies, let alone brute-force molecular dynamics, to address such considerable spatial reorganization of the perforin monomers. In fact, finding suitable collective variables able to track faithfully such large spatial reorganizations is currently not amenable to the available methodology. That said, were we in a position to tackle this daunting challenge, we agree with the Referee, that accounting for the membrane environment would be absolutely necessary. In revision we detail the settings of the simulations (line 539-540, 543, 549-555).

2. The results figures indicate that for each condition, only one simulation run was performed. Is this indeed the case? It is not guaranteed that a single trajectory is representative of the general behavior. A single transition might happen by chance in one condition but not in the other one, even if the differences between them are not significant.

Response: The Reviewer is also correct that for each condition, only one simulation was performed. While there is always an uncertainty that a single trajectory may not be representative of the general behavior, we feel that the length of our molecular dynamics simulations, in excess of 10^6 s, is apposite for the purpose of rationalizing the clockwise conformational transition of PFN2 from pre-pore to pore observed experimentally. We ought to emphasize again that the role of our simulations is not to capture conformational transitions, but merely observe local fluctuations around an average conformation, as well as the evolution of physicochemical properties in response to changes in the environment, like the pH of the aqueous solution. Under these premises, we believe that running additional

simulations would not modify the general conclusions reached from the single, albeit admittedly long trajectories generated for each condition. Last, we would like to remind that the computational assays formed by the perforin trimers in their aqueous environments correspond to nearly 315,000 atoms, the microsecond-timescale simulation of which represents over 25 days of computation on a computer node equipped with 32 cores and two last-generation graphics cards.

3. The CTT domain of CCW subunit-a at pH 4 appears to adopt a distinct (possibly metastable) state (Figure S4) but the paper does not show what this state looks like. It would help to show representative simulation "snapshots" to compare its behavior at pH4 and pH7.

Response: The Reviewer is right in that the CTT domain of subunit a, at pH 4, underwent a partial conformational change, as manifested in the distance root mean-squared deviation plot of Figure S4. We refer to the supporting information of the manuscript, Figure S2 bottom left, that shows what this conformational state looks. While the purpose of this figure is first and foremost to highlight the complementarity of the electrostatic potential, averaged over the length of the simulation and mapped onto the solvent-accessible surface of the different monomers — and, thus, how acidification of the environment is prone to disrupt this complementarity, in particular in a more pronounced way in the clockwise conformational transition than in the anticlockwise — one may, nonetheless, observe how lowering the pH entailed a local conformational transition in the CTT tail of a. In revision, we refer in caption of S4 (line 951-952) to this detail and comment on change in caption of S2 (line 940-941).

4. The fluctuations of CW subunit-c (figure 3) do not appear to vary significantly between pH7 and pH4. It looks like P2 and CTT appear to deviate a bit more from the starting structure (Figure S4). As mentioned in issue 2, in a highly dynamic system, this could be a mere coincidence if observed only in one simulation run (that is relatively short compared to the timescales of the physical process). As in the above point, structural snapshots would be helpful in assessing the conclusions made in the paper.

Response: We agree that there are substantial structural fluctuations apparent in the CW setting at neutral pH7, especially in certain domains of the perforin trimers. Since these fluctuations at pH7 are only found in the CW setting, it appears that it is again specific to this orientation. It is however noteworthy that prepore ring breakages basically never happen — experimentally not observed — at pH7. Indeed, based on Figure 4 (salt bridges) and Figure S2 (charge complementarity at pH7 and pH4), we would not expect that at neutral pH interfaces are fragile. Thus, this observation is somewhat a result of the constraint to simulate a trimer, which is needed to assess the differences at pH4, but maybe not the best setting at pH7. We comment in revision in the caption of Figure S4 (line 952-954).

Minor issues:

- Figure S2: In the caption, the second occurrence of CCW in "CCW (a) and CCW (b)" should be CW, I presume.

Response: The Reviewer is correct — this is, indeed, a typo. We corrected in the revised version of the manuscript from the "CCW (a) and CCW (b)" to "CCW (a) and CW (b)".

Reviewers' Comments:

Reviewer #1:

Remarks to the Author:

The authors have taken on my concerns and made efforts to address these within reason. The manuscript is of high quality overall. The study is a valuable contribution to the field of MPEG1 and MACPF/CDC pore forming proteins, and of course, high-speed AFM imaging of protein dynamics more generally.

This manuscript makes important and insightful advances toward our understanding of an ancient immune effector. Of course, there remain unaddressed questions in the field and this study raises many more.

I look forward to future studies and congratulate the authors on their accomplishment.

Reviewer #3:

Remarks to the Author:

The authors have addressed my questions and concerns in a sufficient manner. Congratulations on this very interesting piece of work!